# Wearable smart sensor systems integrated on soft contact lenses for wireless ocular diagnostics

Joohee Kim[1,*], Minji Kim[1,*], Mi-Sun Lee[1,*], Kukjoo Kim[1], Sangyoon Ji[1], Yun-Tae Kim[2], Jihun Park[1], Kyungmin Na[3], Kwi-Hyun Bae[4], Hong Kyun Kim[5], Franklin Bien[3], Chang Young Lee[2] & Jang-Ung Park[1]

Wearable contact lenses which can monitor physiological parameters have attracted substantial interests due to the capability of direct detection of biomarkers contained in body fluids. However, previously reported contact lens sensors can only monitor a single analyte at a time. Furthermore, such ocular contact lenses generally obstruct the field of vision of the subject. Here, we developed a multifunctional contact lens sensor that alleviates some of these limitations since it was developed on an actual ocular contact lens. It was also designed to monitor glucose within tears, as well as intraocular pressure using the resistance and capacitance of the electronic device. Furthermore, *in-vivo* and *in-vitro* tests using a live rabbit and bovine eyeball demonstrated its reliable operation. Our developed contact lens sensor can measure the glucose level in tear fluid and intraocular pressure simultaneously but yet independently based on different electrical responses.

[1] School of Materials Science and Engineering, School of Energy and Chemical Engineering, Wearable Electronics Research Group, Center for Smart Sensor Systems, Ulsan National Institute of Science and Technology (UNIST), Ulsan 44919, Republic of Korea. [2] School of Life Sciences, School of Energy and Chemical Engineering, Ulsan National Institute of Science and Technology (UNIST), Ulsan 44919, Republic of Korea. [3] School of Electrical and Computer Engineering, Ulsan National Institute of Science and Technology (UNIST), Ulsan 44919, Republic of Korea. [4] Division of Endocrinology, Department of Internal Medicine, Kyungpook National University School of Medicine, Daegu 41944, Republic of Korea. [5] Department of Ophthalmology, Kyungpook National University School of Medicine, Daegu 41944, Republic of Korea. * These authors contributed equally to this work. Correspondence and requests for materials should be addressed to F.B. (email: bien@unist.ac.kr) or to C.Y.L. (email: cylee@unist.ac.kr) or to J.-U.P. (email: jangung@unist.ac.kr).

Wearable electronics are designed to be worn on a person to continuously and intimately monitor an individual's activities, without interrupting or limiting the user's motions[1–8]. Especially, wearable electronics detecting physiological changes for the diagnosis of diseases have recently attracted extensive interests globally[9–13]. Contact lenses, currently worn mainly for vision correction and cosmetic reasons, make continuous contact with our tear fluids and thus provide a unique wearable platform for ocular diagnostics. Electronics on soft contact lenses pose demanding challenges because the system requires reliability upon repeated eye-blinks, flexibility, stretchability and optical transparency for unobstructed vision. Even the most advanced contact lens sensors, however, rely on opaque electronic components constructed on lens-shaped plastic substrates with low oxygen permeability, instead of on actual soft hydrogel lenses, which can limit the safe operation of the devices on a live eye[14–18], as summarized in Supplementary Table 1. Therefore, to overcome these problems and produce a wearable sensor, the substrate should be commercialized contact lens that can be wearing and the materials constituting the sensor and antenna must be transparent, stretchable and harmless to human body.

Among various biomarkers, glucose is particularly important for the diagnosis and management of diabetes. Currently, the finger prick method is commonly used daily by diabetes patients for monitoring the glucose level in blood, but the method accompanies pain and inconvenience during sampling[19]. Instead of detecting the glucose in blood, monitoring the glucose level in other body fluids (for example, urine, saliva, intestinal fluid or tear fluid) may enable noninvasive diagnosis and diabetes control[20]. Tear fluid, in particular, has emerged as a promising body fluid for continuous monitoring of glucose level by recently developed sensor designs[21]. Another significant health indicator that can be obtained from human eyes is the intraocular pressure. Elevated intraocular pressure is the largest risk factor for glaucoma[17,22–24], a leading cause of human blindness. Therefore, early diagnosis and treatment of glaucoma is important, which is however challenging due to the slow and symptomless progression of the disease.

Here, we report transparent and stretchable, multifunctional sensors on wearable soft contact lenses for the wireless detection of glucose and intraocular pressure with high-sensitivity. The key components are graphene and its hybrid with metal nanowires, providing sufficient transparency ($>91\%$) and stretchability ($\sim 25\%$) that ensure reliability, comfort and unobstructed vision when the soft contact lens is worn by users. By integrating the components into a resistance ($R$), inductance ($L$) and capacitance ($C$) circuit operating at a radio frequency, we demonstrate real-time *in-vivo* glucose detection on a rabbit eye and *in-vitro* monitoring of intraocular pressure of a bovine eyeball wirelessly. Power sources, associated circuitry and interconnect electrodes thus are not required in this system. By multiplexing various sensing elements, the contact lens sensors would ultimately enable wireless, continuous and noninvasive monitoring of physiological conditions, as well as the detection of biomarkers associated with ocular and other diseases.

## Results

**Fabrication and characteristics of sensors**. Schematic image of the all-in-one multifunctional sensor composed of a field-effect sensor and antenna on a soft contact lens is given in Fig. 1a. Materials for biosensors operating on soft contact lenses require transparency and stretchability, as well as reliability upon repeated bending and stretching. Candidates satisfying these requirements include graphene[25–28], carbon nanotubes (CNTs)[29,30], metal nanowires (mNWs)[29,31], metal mesh-structures[32,33], conducting polymers[34–36] and their hybrid structures[37–39]. In particular, we previously reported that the graphene-silver nanowire (AgNW) hybrid structure has enhanced electrical and mechanical properties without sacrificing transparency, and is thus suitable as stretchable, transparent electrodes[37]. Here we further verify its reliability by building fully functional integrations of the sensors with circuits on wearable, soft contact lenses using this hybrid (Fig. 1a). The graphene-AgNWs hybrid formed by transferring graphene onto random networks of AgNW (Supplementary Fig. 1). As shown in the picture of the system (Fig. 1b), all the device's components are transparent, with slightly visible spiral antenna (inset). Compared to the single material of graphene or AgNW, the hybrid has significantly reduced sheet resistance (Supplementary Fig. 2) with slightly lower optical transmittance and haziness (Fig. 1c). Furthermore the negligible transconductance of this hybrid structure[37] enables its use as electrodes to build passive electronic components. For example, the hybrid can serve as stretchable and transparent source/drain (S/D) electrodes of a field-effect transistor (FET) with graphene as a channel. This graphene FET (with the hybrid S/D) which are formed on a Si wafer with a 300 nm-thick $SiO_2$ layer shows semi-metallic characteristics with a mobility of $\sim 2,850\,cm^2\,V^{-1}\,s^{-1}$ and Dirac voltage of 30 V (Supplementary Fig. 3). Electrical properties of the device in response to mechanical strain are further investigated for reliable operation on soft contact lenses[40]. The devices on the plastic substrate such as polyethylene terephthalate and polydimethylsiloxane (PDMS) were bent on cylindrical supports with various radii of curvature, completely folded inducing a crease or stretched up to 25% of uniaxial tensile strain (Supplementary Fig. 4). The resulting increase of resistance was negligible ($<10\%$) due to the large elasticity of graphene[41] and the mesh structure of AgNW (ref. 31) (Fig. 1d and e). Resistance of the hybrid remains almost constant ($\Delta R < 6\%$) even after 5,000 cycles of stretching (25% tensile strain) and relaxation, while the value increased to 20% at 10,000 cycles (Fig. 1f). These results suggest that the graphene-AgNW hybrid is a promising component of wearable electronics on soft contact lenses.

**Real-time detection of glucose using graphene sensors**. Among numerous biomolecules included in tear fluid, glucose is an important biomarker for the diagnosis of diabetes. However, the current finger prick method for monitoring the glucose level in blood accompanies pain during blood sampling[19]. Furthermore, it provides only a temporary glucose value, even though continuous monitoring is essential to make an accurate diagnosis. Glucose sensors based on the FET that consists of the graphene channel and hybrid S/D can potentially serve as a pain-free and convenient alternative to the existing approach, especially when integrated onto a wearable contact lens. For selective and sensitive detection of glucose, glucose oxidase (GOD, β-D-glucose from *Aspergillus niger*)[42,43] was immobilized on the graphene channel using a pyrene linker via π–π stacking. Here GOD was attached to the pyrene linker by the amide bond from nucleophilic substitution of N-hydroxysuccinimide[42] (Supplementary Fig. 5). Atomic force microscopy (AFM) images confirm that the GOD selectively binds to the surface of graphene channel as described in Supplementary Fig. 6. The detection mechanism of glucose is illustrated in Fig. 2a. GOD catalyses oxidation of glucose to gluconic acid and reduction of water to hydrogen peroxide. Hydrogen peroxide, a reducing agent in our system, is oxidized to produce oxygen, protons and electrons. The concentration of charge carriers in the channel, and thus the drain current, increases at higher concentration of

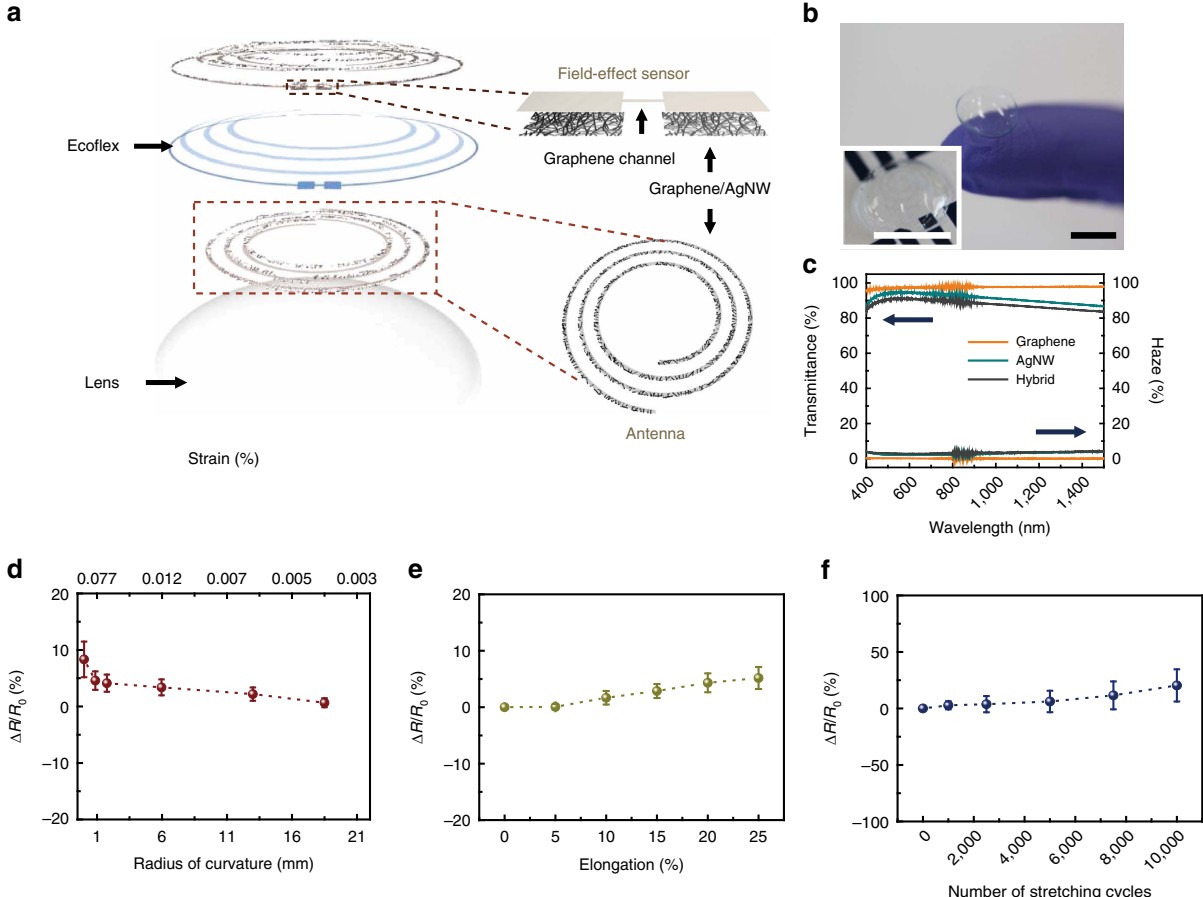

**Figure 1 | Schematic illustration and properties of the wearable contact lens sensor.** (**a**) Schematic of the wearable contact lens sensor, integrating the glucose sensor and intraocular pressure sensor. (**b**) A photograph of the contact lens sensor. Scale bar, 1 cm. (Inset: close-up image of the antenna on the contact lens. Scale bar, 1 cm.) (**c**) Optical transmittance and haze spectra of the bare graphene, AgNWs film and their hybrid structures. (**d**) Relative changes in resistance as a function of outer radius of cylindrical supports (**e**) Relative changes in resistance as a function of tensile strain. (**f**) Relative change in resistance of the graphene FET for 10,000 cycles of stretching and relaxation. Each data point indicates the mean value for 20 samples, and error bars represent the s.d.

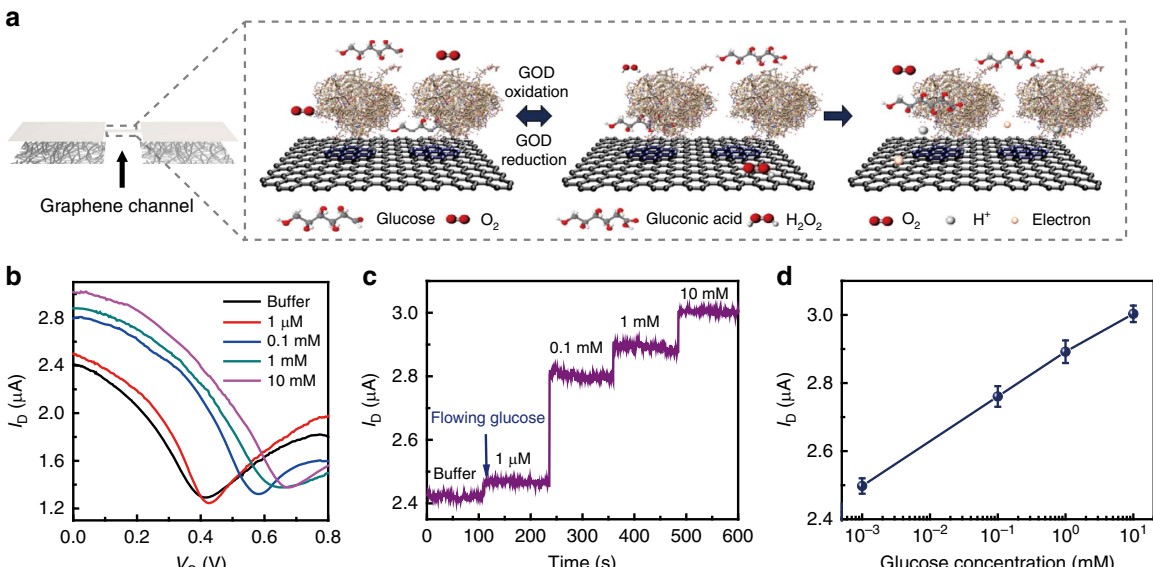

**Figure 2 | Real-time glucose sensing with graphene-AgNW hybrid nanostructures.** (**a**) Schematic illustration and principle of glucose detection with the GOD-pyrene functionalized graphene. (**b**) Transfer ($I_D$-$V_G$) characteristics of the sensor at varied concentrations of glucose ($V_D = 0.1$ V). (**c**) Real-time continuous monitoring of glucose concentrations ($V_G = 0$ V). (**d**) The calibration curve generated by averaging current values and the glucose concentration from 1 μM to 10 mM. Each data point indicates the mean value for 10 samples, and error bars represent the s.d.

glucose[18,44]. As illustrated in Supplementary Fig. 7, we fabricated a block array of 9 FET sensors consisting of graphene for the channel and the hybrid for S/D electrodes and interconnects. The hybrid electrodes and interconnects were passivated with a 500 nm-thick epoxy layer (SU8, Microchem, Inc), except the square-shaped areas which were used for exposing the graphene channels. Here the SU8 as a diffusive barrier lowers molecular concentrations at the already impermeable graphene surface[38], ensuring that no damaging molecule from tear fluid reaches the AgNWs. Grain boundaries in graphene may lower effectiveness of the seal, in particular when the lens is worn for extended periods of time, but the two-layer passivation can provide reasonable protection of the sensor against tear fluids for daily disposable contact lenses. The formation of AgCl, insoluble salts which could be harmful to the human eye, is also prevented by protecting AgNWs from tear fluid which contains chloride ions. Transfer characteristics under various glucose concentrations are given in Fig. 2b. Compared to the buffer-only case (black curve), the drain current increases with glucose concentration due to the positive charge transfer effect of protons ($H^+$) generated from the oxidation of hydrogen peroxide. Based on the transfer characteristic, the drain current under glucose concentrations from 1 µM to 10 mM was measured in real-time at zero gate bias ($V_G = 0$ V) (Fig. 2c). The signal-to-noise ratio (S/N) measured at 1 µM was about 7.34, and the limit of detection at S/N of 3 was 0.4 µM. The device detected glucose concentration of as low as 1 µM, indicating a 10 times improvement over previously reported contact lens sensors made by evaporated metal electrodes[18]. As shown in the calibration curve (Fig. 2d), the sensor was also highly responsive to the typical range of glucose concentrations in tear fluid (0.1–0.6 mM)[18]. Repeating the measurements in artificial tear fluids slightly increased the baseline current but did not degrade the sensitivity (Supplementary Fig. 8). The results confirm that our glucose sensor operates even in the presence of ions and other interfering molecules in tears. We investigated the long-term stability of our sensors with application in a real contact lens in mind. We stored unused sensors in artificial tear solution for up to 24 h, and tested their responses to glucose at varied concentrations (Supplementary Fig. 9). No degradation of the sensitivity after 24 h suggests that the enzymes remain active for at least 24 h. The simple pyrene-chemistry tunes the molecular binding on graphene, and accordingly the multiplexed array of graphene sensors would enable detection of numerous disease-related biomarkers in tear fluid. Although precise diagnosis of glucose may require further development of the sensor, the contact lens sensor can be sufficient for screening prediabetes and daily monitoring of the glucose level.

***In-vivo* test of wireless monitoring of glucose**. In addition to the transparency and stretchability discussed above, the contact-lens device should also have a high-oxygen and water permeability to be compatible with the wearable soft contact lens, instead of lens-shaped polyethylene terephthalate or PDMS substrates[14–18]. Furthermore, since connecting wires to the lens device is not practical, both powering the device and recording the sensor response should be performed wirelessly[45]. Figure 3a illustrates a schematic diagram of the device attached to a soft contact lens in which the graphene-AgNW electrodes and the graphene channel are lithographically patterned on an ultra-thin parylene substrate (~500 nm-thick). Parylene was chosen as substrates instead of other plastic materials due to its intraocular biocompatibility and mechanical superiority such as strength and stretchability[46]. Also, high transparency and conformal pinhole-free deposition make it an ideal substrate for electrical components on the contact lens.

All the components of the device are transparent, with slightly visible spiral antenna, and conformably wrap the curved surface of the contact lens (radius of curvature ~1.4 cm, thickness ~85 µm). The sensor can be modelled as an electrical RLC resonant circuit, comprised of the resistance (R) of the graphene channel, the inductance (L) of the antenna coil made of the graphene-AgNW hybrid and the capacitance (C) of graphene-AgNW hybrid S/D electrodes. A wireless operation can be achieved by mutually coupling the sensor with an external antenna, as described in Fig. 3b. These circuits are connected via a magnetic field, which can be characterized by a coupling coefficient[47,48]. Therefore, the wireless sensing antenna analyses how the reflection condition depends on the resistivity change of the sensor. At varied glucose concentrations, a reflection value (S11) of the wireless sensor was measured at the resonant frequency of 4.1 GHz (Fig. 3c). The reflection was enlarged at higher glucose concentrations, caused by reduced resistance of the graphene upon glucose binding (Fig. 3c; Supplementary Fig. 10). The sensor responds specifically to glucose even in the presence of interferents (50 µM of ascorbic acid, 10 mM of lactate and 10 mM of uric acid) in the tear (Supplementary Fig. 11). Also, these reflection values of the sensor almost accord with the simulation results (Supplementary Fig. 12; Supplementary text). Figure 3d shows a live rabbit wearing the contact lens sensor for an *in-vivo* recording. For the *in-vivo* test, we put the contact lens in the rabbit eye, gave about 3 h for the rabbit to recover from stress, and fed the rabbit. Considering the delayed increase of the blood glucose after the food intake, we measured the reflection (S11) after 5 h of the rabbit wearing the lens, or ~2 h after feeding. The rabbit showed no sign of abnormal behaviour, and the sensor remained stable during repeated eye-blinks (Supplementary Movie 1). After 5 h, the contact lens sensor detected the glucose concentration of the rabbit, and we wirelessly measured the reflection value of sensors, while the rabbit was wearing the lens. As shown in Fig. 3e, the device on the contact lens still functioned and showed a higher reflection than the value before wearing, presumably because of the glucose binding in tear fluid of the rabbit. Our sensing platform integrated onto the contact lens enables wireless and real-time monitoring of the glucose level in the tear fluid, a technology that potentially replaces the current finger prick method.

***In-vitro* test of wireless monitoring of intraocular pressure**. Utilizing the inductance (L) and capacitance (C) in the RLC circuit adds another mode of detection to the resistance-based glucose sensor discussed above. Here we demonstrate wireless recording of intraocular pressure using the contact lens sensor. Intraocular pressure is the main factor in the pathogenesis of glaucoma[17], which eventually leads to the loss of vision. Although the intraocular pressure peak occurs at night rather than daytime, it is frequently measured when patients visit the hospital during daytime due to the time and place limitations, which therefore accompanies possibility of misdiagnosis. Beyond this limitation, in order to continuously measure the intraocular pressure for 24 h, a wearable and transparent intraocular pressure monitoring sensor on a soft contact lens may be one feasible solution. To monitor intraocular pressure, we placed a layer of silicone elastomer (ecoflex) between the two inductive spirals made of graphene-AgNW hybrid electrodes in a sandwich structure. Figure 4a illustrates how the contact lens sensor responds to a raised intraocular pressure, termed ocular hypertension. High-intraocular pressure increases the corneal radius of curvature, which in turn increases both the capacitance by thinning the dielectric and the inductance by bi-axial lateral expansion of the spiral coils. As a result, ocular hypertension shifts the reflection

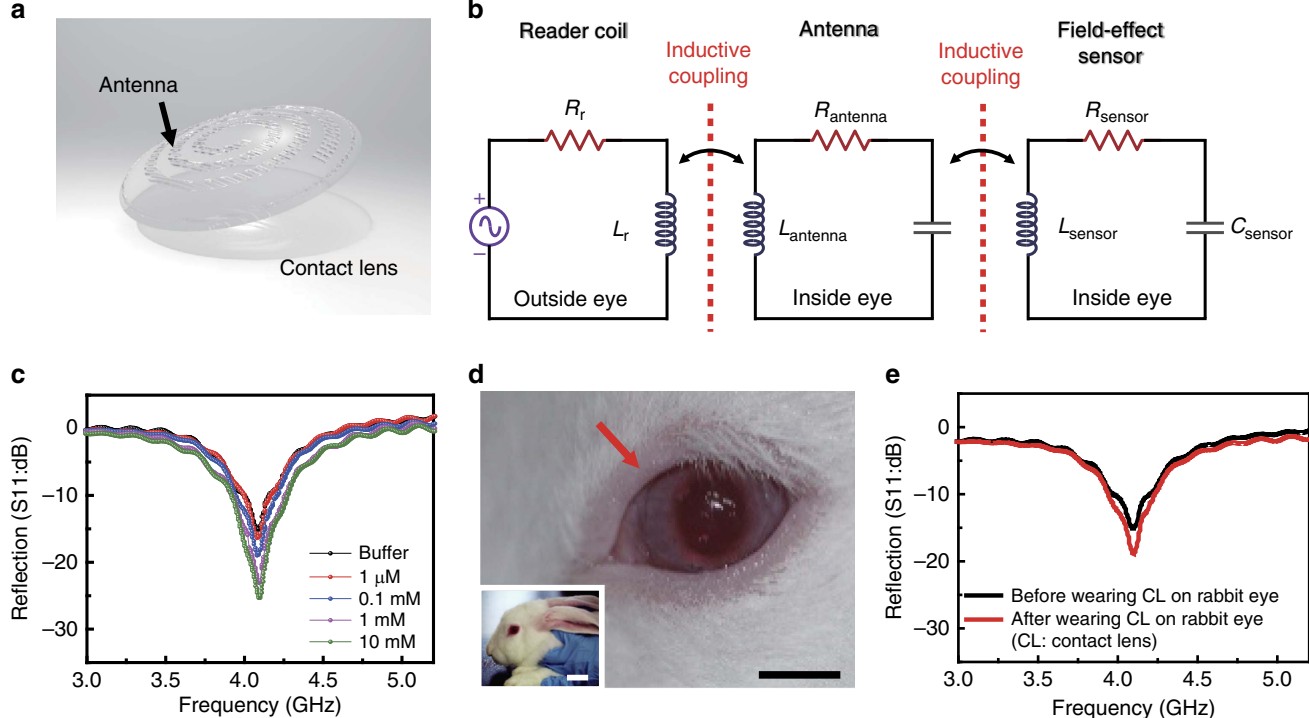

**Figure 3 | Contact lens sensor for wireless detection of glucose.** (**a**) Schematic illustration of the transparent glucose sensor on contact lens. (**b**) Schematic of reading circuit for wireless sensing on contact lens. (**c**) Wireless monitoring of glucose concentrations from 1 μM to 10 mM. (**d**) Photographs of wireless sensor integrated onto the eyes of a live rabbit. Black and white scale bars, 1 cm and 5 cm, respectively. (**e**) Wireless sensing curves of glucose concentration before and after wearing contact lens on an eye of live rabbit.

spectra of the spiral antenna to a lower frequency[10]. For *in-situ* wireless intraocular pressure sensing, the external reader coil was aligned over the contact lens sensor along the same axis (Fig. 4b). The readout system was able to detect the resonance frequency, $f_{sensor}$, a function of inductance and capacitance as described in the Methods section[10,17]. The contact lens sensor was tested *in-vitro* on a bovine eyeball because of its structural resemblance to the human eyeball. As presented in Fig. 4c, the sensor on the bovine or mannequin eye has sufficient transparency without obstructing the field of vision. Figure 4d shows the reflection spectra wirelessly collected from the contact lens sensor worn by the bovine eyeball. The value for $f_{sensor}$ down-shifted at higher intraocular pressure, caused by the raised inductance and capacitance as described above. Here the resonance frequency of the sensor is inversely proportional to the square root of pressure, $f_{sensor} \sim P^{-0.5}$, as shown in Supplementary Fig. 13. In this graph, the frequency response is almost linear for relatively small pressure (below 50 mm Hg). In the physiologically relevant range of intraocular pressure, 5–50 mm Hg (ref. 17), the $f_{sensor}$ decreased linearly with pressure by the slope of 2.64 MHz mm Hg$^{-1}$ (Fig. 4e). Also Supplementary Fig. 14 shows a linear relationship of the relative capacitance change by pressure for this intraocular pressure range ($C \sim P$). As shown in Fig. 4f, the frequency response to intraocular pressure is reproducible with negligible hysteresis. Measured pressures were highly correlated with the real intraocular pressures, which were examined by a pressure sensor inserted into the eyeball. The frequency responses were consistent even when the device slipped to different locations on the eyeball (Supplementary Fig. 15). This is because of the layer that prevents the active components of the device from making direct contact with the eyeball. As the two sensing modes in principle operate independently of each other, the sensors can be potentially multiplexed for simultaneous detection of glucose and intraocular pressure. This wearable smart contact lens will be a promising application in wireless and real-time ocular diagnostics without obstruction to vision.

## Discussion

We have demonstrated a wearable smart contact lens with highly transparent and stretchable sensors that continuously and wirelessly monitors glucose and intraocular pressure, which are the risk factors associated with diabetes and glaucoma, respectively.

In conclusion, compared to the existing contact lens sensors made of conventional opaque materials, the breakthrough was made by the hybrid structure of 1D and 2D nanomaterials, which adds reliability and robustness to the high conductivity, flexibility and transparency of each material. Among the three elements in the demonstrated RLC circuit, $R$ responds to molecular binding, whereas $L$ and $C$ vary with structural changes of the device, thus enabling the detection of intraocular pressure. Here, the simple pyrene-chemistry allows selective binding of target biomarkers onto graphene, which can be tuned for a wide range of analytes. Furthermore, the change of reflection coefficient by $R$, and the shift of resonance frequency by $L$ and $C$ operates independently of each other. Therefore, this multiplexed contact lens sensor indicates substantial promise for next-generation ocular diagnostics, which not only monitors disease-related biomarkers but also evaluates ocular and overall health conditions of our body.

## Methods

**Preparation of AgNW films.** AgNWs (Nanopyxis Co. Ltd,) with average diameter of 30 ($\pm$ 5) nm and length of 25 ($\pm$ 5) μm were dispersed in deionized water (3 mg ml$^{-1}$). The dispersion was spin-coated on a target substrate for 30 s at 500 r.p.m. to get the lowest sheet resistance. The substrate was then annealed at 150 °C for 90 s to completely evaporate the water.

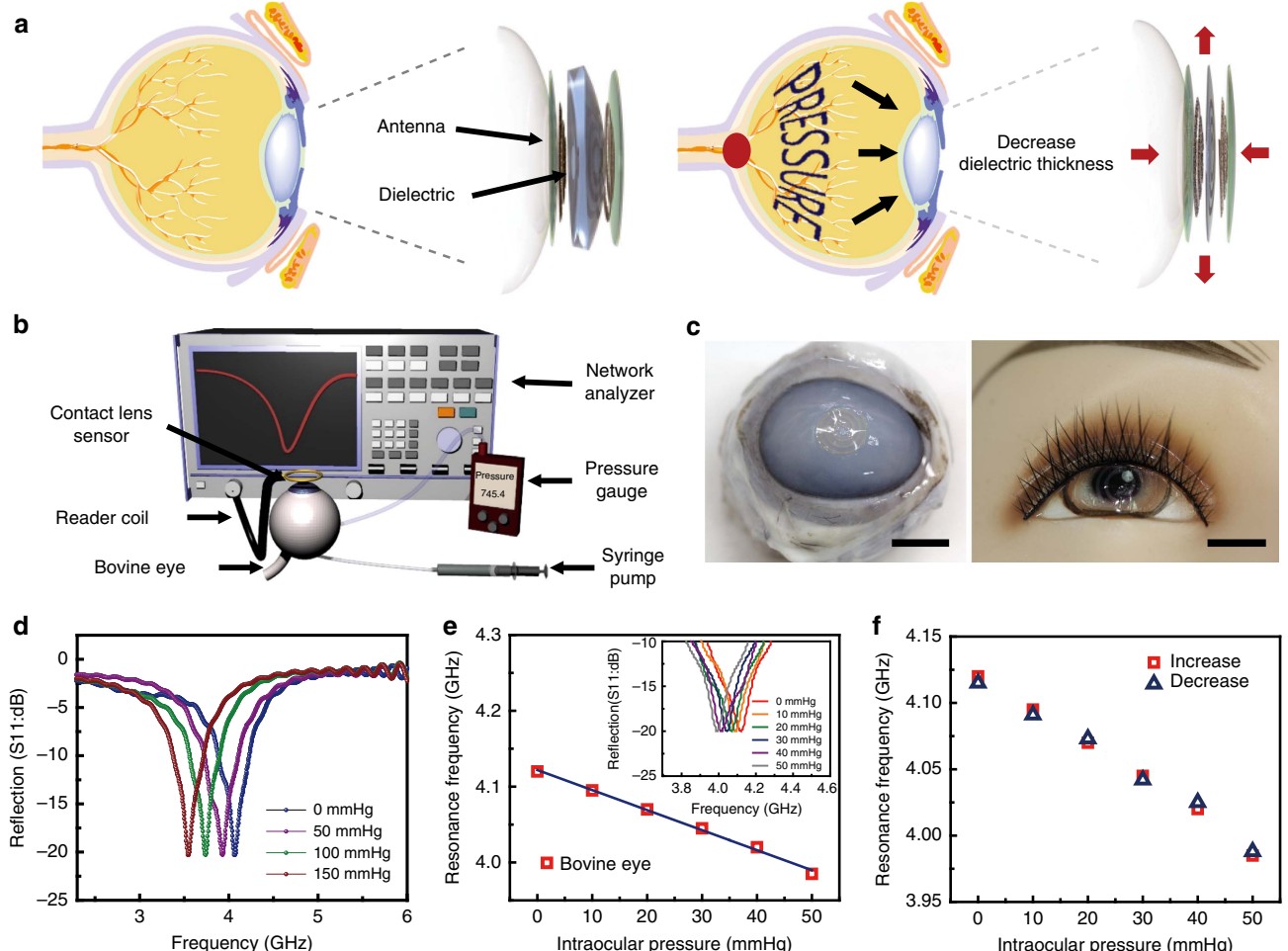

**Figure 4 | Contact lens sensor for wireless monitoring of intraocular pressure.** (**a**) Schematic showing the mechanism of intraocular pressure sensing. (**b**) Schematic of the experimental set-up for wireless intraocular pressure sensing. (**c**) Photographs of the sensor transferred onto the contact lens worn by a bovine eyeball (left) and a mannequin eye (right). Scale bar, 1 cm. (**d**) Wireless recording of the reflection coefficients at different pressures. (**e**) Frequency response of the intraocular pressure sensor on the bovine eye from 5 mm Hg to 50 mmHg. (Inset: the corresponding reflection coefficients of the sensor) (**f**) Frequency response of the sensor during a pressure cycle.

**CVD synthesis and transfer of graphene.** A Cu foil (Alfa Aeasr, item No.: 13382) cleaned using acetone, IPA and deionized (DI) water was loaded into the CVD chamber. After pumping the chamber down to 10 mtorr, the furnace was heated up to 1,000 °C under 200 s.c.c.m. Ar and 500 s.c.c.m. $H_2$. Graphene synthesis was carried out for 5 min under 12 s.c.c.m. of $CH_4$ and 500 s.c.c.m. of $H_2$. Then the chamber was cooled down to room temperature with Ar flowing at 500 s.c.c.m. To transfer the synthesized graphene onto a target substrate poly (methyl methacrylate) (MicroChem Corp., 950 PMMA) was spin-coated on the graphene-on-Cu foil. Floating the foil on a diluted etchant (FeCl₃: HCl: $H_2$ = 1:1:20 vol%) completely removed the foil, and a layer of PMMA-coated graphene remained. After rinsing the graphene-PMMA with DI water, the layer was transferred onto a target substrate, followed by removal of PMMA by acetone.

**Fabrication of field-effect transistors.** Photoresist was patterned on spin-coating AgNW films, and AgNW electrodes were made by removing the unprotected AgNW films via both dry-etching using oxygen plasma in reactive ion etching (RIE) (50 W, 60 s.c.c.m., 240 s) and wet-etching using $H_3PO_4$: $C_2H_4O_2$: $C_6H_4NO_5SNa$: $H_2O$ (55:1:4:40 vol%) for 10 s. After removing the PR in acetone, graphene was transferred onto the AgNW electrodes and patterned by photolithography and RIE for the hybrid electrodes and channel. The width and length of the channel were 5 and 50 μm, respectively.

**Electrical characterization.** We used a probe station (Keithley 4200-SCS semiconductor parametric analyser) to measure sheet resistance, and transfer and output characteristics of devices (back- and solution- gated characterizations). In solution-gated characterization, we used Ag/AgCl wire as a reference electrode. All devices were characterized at 0.1 V of drain bias.

**Optical characterization.** Optical transmittance measurements were conducted by ultraviolet–vis-NIR spectroscopy (Cary 5000 UV-vis-NIR, Agilent) with the transmittance of the substrate subtracted. Electron micrographs of our samples were obtained with scanning electron microscope (SEM, Hitachi, S-4800).

**Immobilization of glucose oxidase.** The sensor was immersed in 1 mg ml$^{-1}$ solution of 1-pyrenebutanoic acid, succinimidyl ester (Sigma-Aldrich, USA) in methanol for 2 h at room temperature and further cleaned with methanol. The sensor was then placed in 10 mg ml$^{-1}$ GOD (Sigma-Aldrich, USA) in deionized water for 18 h at room temperature, rinsed with DI water and dried with nitrogen gas.

**Microfluidic platform for real-time glucose sensing.** Field-effect transistor (FET)-type sensor was fabricated using graphene-AgNW hybrid as S/D electrodes, graphene as an active channel layer, Cr/Au as interconnect and SU8 as a passivation layer. This passivation layer was a square of side 40 μm. A 5 mm- thick PDMS block (Sylgard 184 silicone elastomer: curing agent = 10:1 wt%), which contains a micro-fluidic channel, was punched out for the inlet, outlet and gate terminal, and then Ag/AgCl wire as a reference electrode was inserted into the channel. After that, the PDMS block was physically fixed on the sensor. A buffer solution (Samchun Pure Chemical Co., pH 7.00 ± 0.02) containing glucose at various concentrations was passed through the microfluidic channel at a constant rate (1 ml h$^{-1}$) with a syringe pump (New Era Pump Systems, Inc., NE-300). The sensor was characterized with a semiconductor parameter analyser (Keithley 4200-SCS). For real-time sensing the drain current was recorded versus time at fixed drain voltage ($V_D$) of 0.1 V and gate voltage ($V_G$) of 0 V.

**Fabrication of intraocular pressure sensor.** A Cu foil was coated with parylene (~500 nm-thick) using a parylene coater (Alpha plus). AgNW suspension was

spin-coated and annealed on the parylene substrate, and AgNW spiral coil was patterned by conventional etch-back process using photo-resist and reactive ion etching process. Then Ecoflex (Smooth-On Ecoflex 0030) was spin-coated above the bottom AgNW spiral coil. Afterwards, top AgNW spiral coil was formed by the same methods as the bottom coil, followed by passivation of the device by another layer of parylene. The bottom Cu foil was etched by Ni etchant, and the device was transferred onto a contact lens. Finally, the central area of the sensor was punched to allow permeation of oxygen and water.

**Wireless sensing measurement.** The wireless sensor was designed to have an RLC passive circuit. The helix coils were made of an electrically conducting and optically transparent graphene-AgNW hybrid including three turn helix in an outer coil with a width of 500 μm and a single turn helix in an inner coil with a width of 120 μm. The wireless sensor was tested by a network analyser (Rohde&Schwarz, znb 8) at various glucose concentrations from 1 μM to 10 mM and at intraocular pressures ranging from 5 to 50 mm Hg. A reader antenna was used to inductively couple and power the remote sensor. The distance between the contact lens sensors and the reading antenna was 10 mm. The resistance change of graphene in response to glucose causes changes in the reflection value (S11 parameter) at resonance frequency. The resonance frequency is shifted by the change of capacitance with the same reflection value by following the equation. By using the Kirchhoff's circuit laws, the S11 is related to the channel resistance ($Z_3$) by equation (1),

$$S11 = \frac{\omega^2 M_{12}^2}{2Z_1 Z_2 + \frac{2\omega^2 M_{12}^2 Z_1}{Z_3} + \omega^2 M_{12}^2} \tag{1}$$

where $M_{xy}$, and $Z_x$ are the coupling coefficient, and resistance of reader coil (1), resonate receiver coil (2) and load coil (3), respectively.

The resonance frequency is shifted by the change of capacitance with the same reflection value by following the equation (2).

$$f_{sensor} = \frac{\omega_{sensor}}{2\pi} \approx \frac{1}{2\pi\sqrt{L_{eff}C_{eff}}} \left( \text{if } R^2 \ll \frac{L}{C} \right) \tag{2}$$

**Rabbit experiment.** For *in-vivo* glucose sensing we used a male New Zealand white rabbit. The test was performed according to the guidelines of the National Institutes of Health for the care and use of laboratory animals, and with the approval of the Institute of Animal Care and Use Committee of UNIST (UNI-STIACUC-14-024). Institute of Animal Care and Use Committee of UNIST is ethics review committee.

**Selectivity test.** For the selectivity test, the sensor is tested by the solution of 0.1 mM glucose, 50 μM ascorbic acid (Product #PHR1008, Sigma-Aldrich, USA), 10 mM lactate (Product #PHR1113, Sigma-Aldrich, USA) and 10 mM urea (Product #PHR1406, Sigma-Aldrich, USA).

**Bovine eyeball experiment.** The *in-vitro* test was performed using the bovine eyeball. The intraocular pressure of the eyeball was measured by a pressure sensor (Testo 511) inserted into the eye chamber, and also controlled by a syringe pump with a needle inserted into the eyeball. The change of resonance frequency was characterized wirelessly using the network analyser (Rohde&Schwarz, znb 8).

**Data availability.** Data supporting the findings of this study are available within the article and its supplementary information files and from the corresponding author on reasonable request.

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

## Acknowledgements

This work was supported by the Ministry of Science, ICT & Future Planning and the Ministry of Trade, Industry and Energy (MOTIE) of Korea through the National Research Foundation (2016R1A2B3013592 and 2016R1A5A1009926), the Technology Innovation Program (Grant 10044410), the Nano Material Technology Development Program (2015M3A7B4050308 and 2016M3A7B4910635), the Convergence Technology Development Program for Bionic Arm (NRF-2014M3C1B2048198), the Pioneer Research Center Program (NRF-2014M3C1A3001208), the Human Resource Training Program for Regional Innovation and Creativity (NRF-2014H1C1A1073051). Also, the authors thank CooperVision Awards and financial support by the Development Program of Manufacturing Technology for Flexible Electronics with High Performance (SC0970) funded by the Korea Institute of Machinery and Materials, and by the Development Program of Internet of Nature System (1.150090.01) funded by UNIST.

## Author contributions

J.K., M.K. and M.-S.L. contributed equally to this work. J.K., M.K. and M.-S.L. designed and performed the experiments, fabricated the devices and analysed the data. K.K., S.J. and J.P. contributed to the sample preparations and device data analysis. Y.-T.K. synthesized glucose and glucose oxidase compounds. K.N. and F.B. designed and simulated the antenna. H.K.K. and K.-H.B. performed analysis. J.-U.P. and C.Y.L. oversaw all research phases and revised the manuscript. All authors discussed and commented on the manuscript.
