## [Peer Review File · Nature Communications]

Reviewer #1 (Remarks to the Author):

The manuscript "Wearable Smart Sensor...", by Kim et al describes an interesting platform technology that allows high quality chemical and physical sensing using devices integrated directly with contact lenses, in formats and with materials that do not obscure normal vision. The results include not only device demonstrations but also illustrations of use on live animal models. These advances are quite impressive, and well suited for publication in a top journal such as Nature Communications. I can give it a positive recommendation, but only after the authors address the following comments in revisions:

- 1) The authors mention the use of SU8 as a layer to protect the AgNWs from tear fluid, yet it is well known that polymers cannot provide robust water barriers. The authors should present some data on the timescales for diffusion of water through a 500 nm thick layer of SU8 - I suspect water penetration within a few days, maybe faster. Some discussion should be included.
- 2) The measurement capabilities for glucose and pressure are interesting, but the authors must provide quantitative comparisons of their in vitro and in vivo measurements against clinical gold standards, in order to establish validity.
- 3) The authors refer to the use of parylene, yet this material is not stretchable. Some clarification is necessary.
- 4) For measurement of glucose, the authors should report any other species present in tears that could interfere with the readings.
- 5) In measurements what kinds of parasitics could influence the measurements. For instance, if the moisture of the eye, or the volume of tear fluids changes, then what is the impact on the wireless reading?
- 6) The illustration in Fig. 4 suggests that the pressure reading might depend strongly on interface slippage between the contact lens and the surface of the eye. Some discussion is necessary.
- 7) In terms of referencing, I feel that the authors do not do an adequate job in delineating the origins and the most sophisticated forms of stretchable electronics. Representative citations from the Princeton, Illinois and Tokyo groups would seem appropriate, to provide the reader with a better perspective on the field.

Reviewer #2 (Remarks to the Author):

This is an interesting paper describing a novel technology for measuring glucose levels and IOP through a contact lens based approach.

The study is well designed and conducted. The manuscript well written.

The authors should, however, acknowledge previous work on the first "smart" contact lens for measuring IOP by including following references:

1. First steps toward noninvasive intraocular pressure monitoring with a sensing contact lens. Leonardi M, Leuenberger P, Bertrand D, Bertsch A, Renaud P. Invest Ophthalmol Vis Sci. 2004 Sep;45(9):3113-7.
2. Continuous intraocular pressure monitoring with a wireless ocular telemetry sensor: initial clinical experience in patients with open angle glaucoma. Mansouri K, Shaarawy T. Br J Ophthalmol. 2011 May;95(5):627-9. doi: 10.1136/bjo.2010.192922. Epub 2011 Jan 7.
3. Continuous 24-hour monitoring of intraocular pressure patterns with a contact lens sensor: safety, tolerability, and reproducibility in patients with glaucoma. Mansouri K, Medeiros FA, Tafreshi A, Weinreb RN.

Reviewer #3 (Remarks to the Author):

This paper reported a multifunctional sensor that can be attached to soft contact lens based on a RLC circuit. R (resistance) responds to binding of glucose in tear fluid while C (capacitance) change in accordance with structural changes of the contact lens induced by varying intraocular pressure. The work is interesting. However, the work is quite qualitative and substantial analyses are needed. Major revision is needed before considering for publication.

Below are several detailed comments:

1. The glucose sensing is based on a field-effect sensor. As shown in Fig. 1a or 2a, the sensor consists of source/drain electrodes and channel (graphene), but there is no gate electrode. It's unclear how the authors adjusted the gate voltage.
2. The overall sensor structure and the RLC mechanism are not clear. For instance, it's unclear how the glucose sensor is connected to the antenna, through inductive coupling as in Fig. 3b. I understand the inductive coupling between the external reader and the antenna, but don't understand why there is another inductive coupling between the sensor and the antenna (both are internal components).
3. The authors showed that the intraocular pressure can shift the resonance frequency of the external reader. The frequency shift is due to the change in C and L, as a result of the deformation of the eyeball. It is important to measure the deformation as a function of the applied pressure and perform some analyses to correlate (and confirm) the frequency shift is indeed due to the eyeball deformation.

Reviewer #4 (Remarks to the Author):

The paper reported the preparation of flexible smart sensors that can be used for testing tears and intraocular pressure. The devices are simple and the sensing mechanisms are similar to those in literature (Nanoscale, 2010,2, 1485-1488). So the novelty of this work lies on the integration of the test systems with contact lens. However, the devices have too many problems for practical applications, such as the specificity of the sensing process. It is unclear whether other biomaterials (e.g. proteins, DNA, ascorbic acid and uric acid) will also influence the performance of the graphene transistor. On the other hand, the stress may influence the conductivity of graphene film since graphene is stress sensitive (Nat. Comm. 2:255, DOI: 10.1038/ncomms1247), which will influence the results for glucose tests as well. So the obtained signal cannot be attributed to the change of glucose concentration only. Moreover, there are many technical problems as follows should be clarified:

- (1) The sensitivity of a biosensor is limited by signal to noise ratio. What is the signal level of the devices in Figure 2-4. It seems that the curves in Fig 3c and 4d are smoothed by data treatment. It is necessary to show original data in the figure without smoothing.
- (2) The resistance change in graphene transistor in glucose solution is ~20%. Theoretical calculation should be provided to fit the change of reflection in Figure 3c and indicate that the obtained glucose concentrations are consistent with the results in Figure 2b.
- (3) Although claimed by the authors, the device may be not suitable for long-term test (up to several hours) of glucose concentration since the enzyme may lose its activity very quickly. Control experiments should be provided to show the long term stability of the glucose sensors.
- (4) How about the influence of salt on the conductivity of graphene in tears? The change of signal before and after being worn on eyes in figure 3e is possibly due to the influence of salt in tears.

In summary, the devices are not new and the integrated systems may be useful for some practical applications. However, the major problem of the systems is the reliability and selectivity in sensing processes, which should be clarified. So the paper is not suitable for publication at this stage.

[Response to Reviewer #1]

We thank the reviewer for a thoughtful review of our manuscript, and we welcome the opportunity to address and clarify the issues raised in the reviewer report. Our responses are as follows:

Comment 1: “The authors mention the use of SU8 as a layer to protect the AgNWs from tear fluid, yet it is well known that polymers cannot provide robust water barriers. The authors should present some data on the timescales for diffusion of water through a 500 nm thick layer of SU8 - I suspect water penetration within a few days, maybe faster. Some discussion should be included.”

Response to Comment 1:

We would like to thank the reviewer for bringing this point to our attention. Our devices have the AgNWs(bottom) / graphene(top) hybrid structure. As oxygen gas and moisture cannot pass through graphene¹, the top graphene layer alone is efficient at blocking oxygen and water. The additional layer of SU8, which serves as a diffusional barrier that creates a concentration gradient, further ensures protection of the AgNWs from tear fluid. We have previously shown that this graphene-SU8 layer effectively protects the underlying AgNWs². We modified the manuscript to address this stability issue.

References

- 1 Bunch, J. S. *et al.* Impermeable atomic membranes from graphene sheets. *Nano. Lett.* **8**, 2458-2462 (2008).
- 2 Lee, M.-S. *et al.* High-performance, transparent, and stretchable electrodes using graphene–metal nanowire hybrid structures. *Nano Lett.* **13**, 2814-2821 (2013).

Revised Manuscript (page 6):

The hybrid electrodes and interconnects were passivated with a 500 nm-thick SU8 layer, except the square-shaped areas which were used for exposing the graphene channels. **Here the SU8 as a diffusive barrier lowers molecular concentrations at the already impermeable graphene surface³⁸, ensuring that no damaging molecule from tear fluid reaches the AgNWs.** The formation of AgCl, insoluble salts which could be harmful to the human eye, is **also** prevented by protecting AgNWs from tear fluid which contains chloride ions.

Comment 2: “The measurement capabilities for glucose and pressure are interesting, but the authors must provide quantitative comparisons of their *in vitro* and *in vivo* measurements against clinical gold standards, in order to establish validity.”

Response to Comment 2:

We would like to thank the reviewer for bringing this point to our attention. In the case of glucose sensor (*in-vivo* test), tear glucose in diabetic patients has been studied for over 80 years since Michail *et al.* first reported the approach¹. Finding correlation between the glucose level in tear and in blood is highly promising as supported by many recent studies on the contact-lens-based glucose sensors²⁻⁶. Finding the exact correlation between the two, however, requires the development of unambiguous sampling methods, and thus is beyond the scope of our study. Therefore, we cannot compare our measurements with clinical gold standards because the clinical gold standard of measuring glucose level in tear are until ambiguous.

In the case of intraocular pressure sensor (*in-vitro* test), applanation tonometry is considered to be the gold standard for measuring intraocular pressure (IOP) in clinical settings. The IOP is inferred from the amount of force required to flatten a constant area of the cornea. We performed *in-vitro* test using bovine eyeballs. The applanation tonometry, which needs to be directly applied to cornea, cannot be performed when the contact lens is worn. In our study intraocular pressure was measured by a pressure sensor (TESTO 511) inserted into the eye. Although the intraocularly inserted sensor cannot be used in clinical settings, the results are very reproducible and highly correlated with the real IOP.

Reference

- 1 Michail, D., Vancea, P. & Zolog, N. Lacrimal elimination of glucose in diabetic patients. *C. R. Soc. Biol. Paris.* **125**, 194-195 (1937).
- 2 Alexeev, V. L., Das, S., Finegold, D. N. & Asher, S. A. Photonic crystal glucose-sensing material for noninvasive monitoring of glucose in tear fluid. *Clinical Chemistry* **50**, 2353-2360 (2004).

- 3 Badugu, R., Lakowicz, J. R. & Geddes, C. D. Ophthalmic glucose monitoring using disposable contact lenses—a review. *Journal of fluorescence* **14**, 617-633 (2004).
- 4 March, W. F., Mueller, A. & Herbrechtsmeier, P. Clinical trial of a noninvasive contact lens glucose sensor. *Diabetes technology & therapeutics* **6**, 782-789 (2004).
- 5 Domschke, A., March, W. F., Kabilan, S. & Lowe, C. Initial clinical testing of a holographic non-invasive contact lens glucose sensor. *Diabetes technology & therapeutics* **8**, 89-93 (2006).
- 6 Ascaso, F. J. & Huerva, V. Noninvasive Continuous Monitoring of Tear Glucose Using Glucose-Sensing Contact Lenses. *Optometry and vision science: official publication of the American Academy of Optometry*.

Revised Manuscript (page 10):

As shown in Fig. 4f, the frequency response to intraocular pressure is reproducible with negligible hysteresis. **Measured pressures were highly correlated with the real intraocular pressures, which were examined by a pressure sensor inserted into the eyeball.**

Comment 3: “The authors refer to the use of parylene, yet this material is not stretchable. Some clarification is necessary.”

Response to Comment 3:

We thank the referee for this comment. We adopted parylene C as a substrate for assembling electronic components, and then transferred the whole system onto an actual contact lens in its wet swollen state. The dimensional changes of sensor between flat and convex state is negligible because the yield elongation of parylene C is 2.9%¹. Also this material can stand 200% stretching until the break¹. Therefore, we believe that parylene C are an ideal substrate for electrical components on the contact lens with other advantages of parylene, such as high transparency and conformal pinhole-free deposition.

References

- 1 Specialty coating systems, Inc. Parylene information sheets.
<http://www.nbtc.cornell.edu/sites/default/files/Parylene%20Information%20Sheets.pdf>

Comment 4: “For measurement of glucose, the authors should report any other species present in tears that could interfere with the readings.”

Response to Comment 4:

As the reviewer correctly pointed out, the ions and other molecular species in tear fluid can indeed affect our measurements. We thus performed additional experiments to study the effect of other species in tears. Supplementary Figure 8 below shows the sensor responses to varied concentrations of glucose in buffer (blue) and in artificial tears (red). In both cases, we were able to detect glucose from 1 μM to 10 mM. The slight increase in the baseline current in tears did not degrade the sensitivity. The results confirm that our glucose sensor operates even in the presence of ions and other interfering molecules in tears. We have added the results and discussion to the revised manuscript.

Revised Manuscript (page 7):

The device detected glucose concentration of as low as 1 μM , indicating a 10 times improvement over previously reported contact lens sensors made by evaporated metal electrodes¹⁸. As shown in the calibration curve (Fig. 2d), the sensor was also highly responsive to the typical range of glucose concentrations in tear fluid (0.1 ~ 0.6 mM)¹⁸. **Repeating the measurements in artificial tear fluids slightly increased the baseline current but did not degrade the sensitivity (Supplementary Figure 8). The results confirm that our glucose sensor operates even in the presence of ions and other interfering molecules in tears.**

Revised Supplementary Information (page 8):

Supplementary Figure 8 | Detection of glucose in different solution environments. The current characteristics of the sensor in response to the glucose dissolved in buffer (blue) and artificial tears (red).

Comment 5: “In measurements what kinds of parasitics could influence the measurements. For instance, if the moisture of the eye, or the volume of tear fluids changes, then what is the impact on the wireless reading?”

Response to Comment 5:

As stated in Supplementary Information, the reflection value (S11 parameter) and the resonance frequency are influenced by the capacitance and inductance. The capacitance is inversely proportional to the thickness of the dielectric layer, so the intraocular pressure can be measured in the capacitance response of the sensor. The inductance of the wireless system can be expressed as

$$L = \frac{\mu_0 \cdot n^2 \cdot D_{avg}}{2} \cdot \left(\ln\left(\frac{2.46}{\sigma}\right) + 0.2 \cdot \sigma^2 \right) \quad (1)$$

where μ_0 is the vacuum permeability; n is the number of antenna turns; σ is the fill ratio, $(D_{out} \text{ (outer diameter of antenna)} - D_{in} \text{ (inner diameter of antenna)}) / (D_{out} + D_{in})$; D_{avg} is the average diameter, $(D_{out} + D_{in})/2$. This equation confirms that the wireless reading is affected only by the number of antenna turns and the dimension of antenna. Therefore, the moisture of the eye and the volume of tear fluids changes cannot influence the sensor performance.

Reference

1 Chen, P.-J., Rodger, D. C., Saati, S., Humayun, M. S. & Tai, Y.-C. Microfabricated implantable parylene-based wireless passive intraocular pressure sensors. *J. Microelectromech. Syst.* **17**, 1342-1351 (2008).

Comment 6: “The illustration in Fig. 4 suggests that the pressure reading might depend strongly on interface slippage between the contact lens and the surface of the eye. Some discussion is necessary.”

Response to Comment 6:

We appreciate the reviewer’s valuable and insightful comment. We performed additional experiments to examine how the slippage affects the intraocular pressure sensor. As shown in Supplementary Fig. 10 below, the frequency responses to the intraocular pressure change were consistent even when the device slipped to different locations on the eyeball. This is because of the layer that prevents the active components of the device from making direct contact with the eyeball. We included these new data in the revised manuscript.

Revised Manuscript (page 10):

As shown in Fig. 4f, the frequency response to intraocular pressure is reproducible with negligible hysteresis. Measured pressures were highly correlated with the real intraocular pressures, which were examined by a pressure sensor inserted into the eyeball. The frequency responses were consistent even when the device slipped to different locations on the eyeball (Supplementary Fig. 11). This is because of the layer that prevents the active components of the device from making direct contact with the eyeball.

Revised Supplementary Information (page 13):

Supplementary Figure 11 | The effect of slippage on the frequency response of the intraocular pressure sensor. Photographs of the intraocular pressure sensor on the (a) left, (b) centre, and (c) right of the bovine eyeball. Scale bars, 1 cm. The frequency response of the sensor on the (d) left, (e) centre, and (f) right of the bovine eyeball at 0 mmHg and 50 mmHg.

Comment 7: “In terms of referencing, I feel that the authors do not do an adequate job in delineating the origins and the most sophisticated forms of stretchable electronics. Representative citations from the Princeton, Illinois and Tokyo groups would seem appropriate, to provide the reader with a better perspective on the field.”

Response to Comment 7:

We would like to thank the reviewer for bringing this point to our attention. We entirely agree with the reviewer’s opinion and included additional comments and relevant references in the revised manuscript.

Revised Manuscript (page 3):

Wearable electronics are designed to be worn on a person to continuously and intimately monitor an individual’s activities, without interrupting or limiting the user’s motions¹⁻⁸. Especially, wearable electronics detecting physiological changes for the diagnosis of diseases have recently attracted extensive interests globally⁹⁻¹³.

[Response to Reviewer #2]

We thank the reviewer for a thoughtful review of and positive comments about our manuscript, and welcome the opportunity to address and clarify the issues raised in the reviewer report. Our responses to the points raised in the report are as follows:

Comment 1: *“This is an interesting paper describing a novel technology for measuring glucose levels and IOP through a contact lens based approach. The study is well designed and conducted. The manuscript well written. The authors should, however, acknowledge previous work on the first "smart" contact lens for measuring IOP by including following references:*

- 1. First steps toward noninvasive intraocular pressure monitoring with a sensing contact lens. Leonardi M, Leuenberger P, Bertrand D, Bertsch A, Renaud P. Invest Ophthalmol Vis Sci. 2004 Sep;45(9):3113-7.*
- 2. Continuous intraocular pressure monitoring with a wireless ocular telemetry sensor: initial clinical experience in patients with open angle glaucoma. Mansouri K, Shaarawy T. Br J Ophthalmol. 2011 May;95(5):627-9. doi: 10.1136/bjo.2010.192922. Epub 2011 Jan 7.*
- 3. Continuous 24-hour monitoring of intraocular pressure patterns with a contact lens sensor: safety, tolerability, and reproducibility in patients with glaucoma. Mansouri K, Medeiros FA, Tafreshi A, Weinreb RN. Arch Ophthalmol. 2012 Dec;130(12):1534-9. doi: 10.1001/archophthalmol.2012.2280.”*

Response to Comment 1:

We would like to thank the reviewer for bringing relevant papers to our attention. We also think those publications would be very helpful to potential readers interested in this field and accordingly added them as References 22-24 in the revised our manuscript.

Revised manuscript (page 3):

Elevated intraocular pressure is the largest risk factor for glaucoma^{17,22-24}, a leading cause of human blindness.

[Response to Reviewer #3]

We thank the reviewer for a thoughtful review of and positive comments about our manuscript, and welcome the opportunity to address and clarify the issues raised in the reviewer report. Our responses to the points raised in the report are as follows:

Comment 1: “The glucose sensing is based on a field-effect sensor. As shown in Fig. 1a or 2a, the sensor consists of source/drain electrodes and channel (graphene), but there is no gate electrode. It's unclear how the authors adjusted the gate voltage.”

Response to Comment 1:

We would like to thank the reviewer for this comment. For the characterization of our device, we obtained the I_D - V_G curves in Fig. 2b and then measured the real-time current in response to the glucose concentration at zero gate bias in Fig. 2c. Therefore, our glucose sensor does not require a gate electrode in the wireless systems.

Comment 2: “The overall sensor structure and the RLC mechanism are not clear. For instance, it's unclear how the glucose sensor is connected to the antenna, through inductive coupling as in Fig. 3b. I understand the inductive coupling between the external reader and the antenna, but don't understand why there is another inductive coupling between the sensor and the antenna (both are internal components).”

Response to Comment 2:

We would like to thank the reviewer for bringing up this point to our attention. The circuit model used to verify the mechanism of the wireless glucose sensing consists of a reader coil, resonant receiver coil (RX coil), and load coil¹. The reader coil is connected to a source and reflection detection circuit. The load coil is connected to a glucose sensor. The RX coils means antenna as shown in Fig. 3b. These circuits are connected via a magnetic field, which can be characterized by a coupling coefficient. Therefore the wireless sensing antenna analyzes how the reflection condition depends on the resistivity change of the sensor. We have emphasized this information in the revised manuscript to clearly explain the RLC mechanism.

Reference

- 1 Na, K. *et al.* Graphene-Based Wireless Environmental Gas Sensor on PET Substrate. *IEEE Sensor Journal* **16**, 5003-5009 (2016).
- 2 Park, J. *et al.* Wearable, wireless gas sensors using highly stretchable and transparent structures of nanowires and graphene. *Nanoscale* **8**, 10591-10597 (2016).

Revised Manuscript (page 8):

A wireless operation can be achieved by mutually coupling the sensor with an external antenna as described in Fig. 3b. **These circuits are connected via a magnetic field, which can be characterized by a coupling coefficient^{47,48}. Therefore, the wireless sensing antenna analyses how the reflection condition depends on the resistivity change of the sensor.**

Comment 3: “The authors showed that the intraocular pressure can shift the resonance frequency of the external reader. The frequency shift is due to the change in C and L, as a result of the deformation of the eyeball. It is important to measure the deformation as a function of the applied pressure and perform some analyses to correlate (and confirm) the frequency shift is indeed due to the eyeball deformation.”

Response to Comment 3:

We thank the reviewer for sharing concerns on the intraocular pressure detection. With all the other conditions fixed, such as the reader coil-to-sensor distance and the position of the antenna, the intraocular pressure was the only variable during the wireless measurement as illustrated in Fig. 4b. Therefore, the measured frequency shift originates solely from the capacitance change associated with the deformed eyeball.

Furthermore, a passive sensor to monitor *in-vivo* intracranial pressure was previously reported¹. Although this opaque and unstretchable sensor was not a contact lens device, the mechanism was similar with our contact lens device. Therefore, we believe that the resonance frequency is indeed changed by the intraocular pressure.

Reference

- 1 Chen, L. Y. *et al.* Continuous wireless pressure monitoring and mapping with ultra-small passive sensors for health monitoring and critical care. *Nat. Commun.* **5**, 5028 (2014).

[Response to Reviewer #4]

We thank the reviewer for a thoughtful review of and positive comments about our manuscript, and welcome the opportunity to address and clarify the issues raised in the reviewer report. Our responses to the points raised in the report are as follows:

Comment 1: “The devices are simple and the sensing mechanisms are similar to those in literature (*Nanoscale*, 2010, 2, 1485-1488).”

Response to Comment 1:

We would like to thank the reviewer for bringing this point to our attention. As the reviewer mentioned, the sensing mechanism has been reported in several papers¹. However, our platform has unique characteristics compared to previous ones.

First of all, none of the existing contact lens sensors are transparent. Further, a platform that measures both glucose concentration and intraocular pressure has not been developed. The contact lens sensor presented in this work is wireless, transparent, can be multiplexed enabling *simultaneous* detection of both glucose and intraocular pressure, and thus is significantly advanced compared to the existing relevant platforms. Furthermore, our sensor adopts a simple channel and antenna structure which reduces overall size of the lens and thus enhances the wearability of the smart contact lens.

This study therefore presents the first sensor platform that is fully compatible with the soft and transparent nature of the contact lens, and suggests a promising strategy toward future electronics. We revised the manuscript and supplementary information as follows:

Reference

1 Huang, Y. *et al.* Nanoelectronic biosensors based on CVD grown graphene. *Nanoscale* **2**, 1485-1488 (2010).

Revised Manuscript (page 3):

Even the most advanced contact lens sensors, however, rely on opaque electronic components constructed on lens-shaped plastic substrates with low oxygen permeability, instead of on actual soft hydrogel lenses, which can limit the safe operation of the devices on a live eye¹⁴⁻¹⁸ as summarized in **Supplementary table 1**.

Revised Supplementary Information (page 14):

Ref.	Sensor	Novelty	Disadvantages
Nanoscale 2 , 1485-1488	Glucose sensor	1) Fabrication of graphene-based glucose sensor	1) Absence of contact lens application 2) Using the opaque metal (Ag paint) 3) Single sensor 4) Physical connection between sensor and measurement equipment
14	Strain gauge sensor	1) Fabrication of light and biocompatible Oxide based TFT sensor	1) Plastic lens-shaped substrate 2) Using the opaque metal (Ti/Au) 3) Single sensor 4) Physical connection between sensor and measurement equipment
15	Glucose sensor	1) Fabrication of In ₂ O ₃ -based FET sensor using simple solution-processing procedure 2) Glucose or pH sensor	1) Transferred on the artificial eye 2) Using the opaque metal (Cr/Au) 3) Physical connection between sensor and measurement equipment
17	Intraocular pressure sensor	1) Fabrication of wireless intraocular pressure sensor	1) Silicone lens-shaped substrate 2) Using the opaque metal (Cu) 3) Single sensor
18	Glucose sensor	1) Fabrication of sensitive glucose sensor by immobilizing glucose oxidase	1) Lens-shaped PET substrate 2) Using the opaque metal (Cu) 3) Single sensor
Our results	Glucose and intraocular pressure sensor	1) Use of commercialized soft contact lenses 2) Fabrication of wireless sensor 3) Fabrication of flexible/stretchable and transparent sensor based on the graphene/AgNW structures 4) Multiplexed enabling simultaneous detection of both glucose and intraocular pressure	

Supplementary Table 1 | Comparison between this work and the previous reports.

Comment 2: “It is unclear whether other biomaterials (e.g. proteins, DNA, ascorbic acid and uric acid) will also influence the performance of the graphene transistor.”

Response to Comment 2:

As the reviewer correctly pointed out, the ions and other molecular species in tear fluid can indeed affect our measurements. We thus performed additional experiments to study the effect of other species in tears. Supplementary Figure 8 below shows the sensor responses to varied concentrations of glucose in buffer (blue) and in artificial tears (red). In both cases, we were able to detect glucose from 1 μM to 10 mM. The slight increase in the baseline current in tears did not degrade the sensitivity. The results confirm that our glucose sensor operates even in the presence of ions and other interfering molecules in tears. We have added the results and discussion to the revised manuscript.

Revised Manuscript (page 7):

The device detected glucose concentration of as low as 1 μM , indicating a 10 times improvement over previously reported contact lens sensors made by evaporated metal electrodes¹⁸. As shown in the calibration curve (Fig. 2d), the sensor was also highly responsive to the typical range of glucose concentrations in tear fluid (0.1 ~ 0.6 mM)¹⁸. Repeating the measurements in artificial tear fluids slightly increased the baseline current but did not degrade the sensitivity (Supplementary Figure 8). The results confirm that our glucose sensor operates even in the presence of ions and other interfering molecules in tears.

Revised Supplementary Information (page 10):

Supplementary Figure 8 | Detection of glucose in different solution environments. The current characteristics of the sensor in response to the glucose dissolved in buffer (blue) and artificial tears (red).

Comment 3: “The stress may influence the conductivity of graphene film since graphene is stress sensitive (Nat. Comm. 2:255, DOI: 10.1038/ncomms1247), which will influence the results for glucose tests as well.”

Response to Comment 3:

We would like to thank the reviewer for this comment. The article describes that the uniaxial stress (0% - 1%) applied to the single crystal graphene along the specific direction of its C-C bonding can shift the G peak in Raman spectra. In our work, the CVD-synthesized graphene has polycrystalline structures, instead of the single crystal, and also the C-C bonding direction of this polycrystal graphene layer is randomly positioned after transferring this graphene on contact lenses. Since the strain direction in our case is arbitrary and random to the C-C bonding, we and other groups observed the negligible change in its resistance by stretching the polycrystal graphene (in the range with no cracking of graphene).

Comment 4: “The sensitivity of a biosensor is limited by signal to noise ratio. What is the signal level of the devices in Figure 2-4. It seems that the curves in Fig 3c and 4d are smoothed by data treatment. It is necessary to show original data in the figure without smoothing.”

Response to Comment 4:

We appreciate the reviewer’s valuable and insightful comment. As the reviewer suggested, we calculated the signal-to-noise-ratio (S/N) and added the data point in the revised Figure 3c and 4d. We already used the original

data without smoothing in Figure 3c and 4d. To demonstrate the original data, we expressed the data using the point and line as below figure.

Figure A. Original data of reflection about glucose and intraocular pressure sensor.

Revised Manuscript (page 7):

Based on the transfer characteristic, the drain current under glucose concentrations from 1 μM to 10 mM was measured in real-time at zero gate bias ($V_G = 0\text{ V}$) (Fig. 2c). The signal-to-noise ratio (S/N) measured at 1 μM was about 7.34, and the limit of detection at S/N of 3 was 0.4 μM .

Revised Manuscript (page 23):

Figure 3 | Contact lens sensor for wireless detection of glucose. (a) Schematic illustration of the transparent glucose sensor on contact lens. (b) Schematic of reading circuit for wireless sensing on contact lens. (c) Wireless monitoring of glucose concentrations from 1 μM to 10 mM. (d) Photographs of wireless sensor integrated onto the eyes of a live rabbit. Black and white scale bars, 5 cm and 1 cm, respectively. (e) Wireless sensing curves of glucose concentration before and after wearing contact lens on an eye of live rabbit.

Revised Manuscript (page 24):

Figure 4 | Contact lens sensor for wireless monitoring of intraocular pressure. (a) Schematic showing the mechanism of intraocular pressure sensing. (b) Schematic of the experimental set-up for wireless intraocular pressure sensing. (c) Photographs of the sensor transferred onto the contact lens worn by a bovine eyeball (left) and a mannequin eye (right). Scale bar, 1 cm. (d) Wireless recording of the reflection coefficients at different pressures. (e) Frequency response of the intraocular pressure sensor on the bovine eye from 5 mmHg to 50 mmHg. (Inset: the corresponding reflection coefficients of the sensor) (f) Frequency response of the sensor during a pressure cycle.

Comment 5: “The resistance change in graphene transistor in glucose solution is ~20%. Theoretical calculation should be provided to fit the change of reflection in Figure 3c and indicate that the obtained glucose concentrations are consistent with the results in Figure 2b.”

Response to Comment 5:

We appreciate the reviewer’s valuable and insightful comment. As the reviewer suggested, we simulated the reflection values at various glucose concentrations using the equation in the Method. Although the shape of the graph is slightly different, the reflection value is almost the same as the Figure 2b. We added this new data to the revised manuscript.

Revised Manuscript (page 8):

The reflection was enlarged at higher glucose concentrations, caused by reduced resistance of the graphene upon glucose binding (Fig. 3c). Also, these reflection values of the sensor almost accord with the simulation results (Supplementary Fig. 10 and Supplementary text).

Revised Manuscript (page 14):

The resonance frequency is shifted by the change of capacitance with the same reflection value by following the equation. By using the Kirchhoff’s circuit laws, the S11 is related to the channel resistance (Z_3) by equation (1),

$$S_{11} = \frac{\omega^2 M_{12}^2}{2Z_1 Z_2 + \frac{2\omega^2 M_{23}^2 Z_1}{Z_3} + \omega^2 M_{12}^2} \quad (1)$$

where M_{xy} , and Z_x are the coupling coefficient, and resistance of reader coil (1), resonate receiver coil (2), and load coil (3), respectively.

Revised Supplementary Information (page 15-16):

Supplementary text

Wireless sensing measurement

The reflection at the resonance frequency is inverse proportion to the electrical resistance in the graphene channel as resistive element of the circuit. The expression for which is, using the Kirchhoff's circuit laws,

$$\begin{bmatrix} I_1 \\ I_2 \\ I_3 \end{bmatrix} = \begin{bmatrix} Z_1 & j\omega M_{12} & 0 \\ j\omega M_{12} & Z_2 & j\omega M_{23} \\ 0 & j\omega M_{23} & Z_3 \end{bmatrix}^{-1} \begin{bmatrix} V_s \\ 0 \\ 0 \end{bmatrix}$$

$$Z_1 = R_{source} + R_r + j \left(\omega L_r - \frac{1}{\omega C_r} \right) \approx R_{source}$$

$$Z_2 = R_{antenna} + j \left(\omega L_{antenna} - \frac{1}{\omega C_{antenna}} \right) \approx R_{antenna}$$

$$Z_3 = R_{sensor} + j \left(\omega L_{sensor} - \frac{1}{\omega C_{sensor}} \right) \approx R_{sensor}$$

where L_x , C_x , M_{xy} , and Z_x (R_x) are the inductance, capacitance, coupling coefficient, and resistance of reader coil (1), resonate receiver coil (2), and load coil (3), respectively.

$$\begin{aligned} V_s &= Z_1 I_1 + j\omega M_{12} I_2 \\ 0 &= j\omega M_{12} I_1 + Z_2 I_2 + j\omega M_{23} I_3 \\ 0 &= j\omega M_{23} I_2 + Z_3 I_3 \rightarrow I_3 = -\frac{j\omega M_{23} I_2}{Z_3} \end{aligned}$$

$$\begin{aligned} 0 &= j\omega M_{12} I_1 + Z_2 I_2 + j\omega M_{23} \left(-\frac{j\omega M_{23} I_2}{Z_3} \right) \\ &= j\omega M_{12} I_1 + Z_2 I_2 + \frac{\omega^2 M_{23}^2 I_2}{Z_3} \\ &= j\omega M_{12} I_1 + \left(Z_2 + \frac{\omega^2 M_{23}^2}{Z_3} \right) I_2 \rightarrow I_2 = \left(\frac{-j\omega M_{12}}{Z_2 + \frac{\omega^2 M_{23}^2}{Z_3}} \right) I_1 \end{aligned}$$

$$\begin{aligned} V_s &= Z_1 I_1 + j\omega M_{12} I_2 = Z_1 I_1 + j\omega M_{12} \left(\frac{-j\omega M_{12}}{Z_2 + \frac{\omega^2 M_{23}^2}{Z_3}} \right) I_1 \\ &= Z_1 I_1 + \left(\frac{\omega^2 M_{12}^2}{Z_2 + \frac{\omega^2 M_{23}^2}{Z_3}} \right) I_1 \end{aligned}$$

$$Z_{in} = \frac{V_s}{I_1} = Z_1 + \frac{\omega^2 M_{12}^2}{Z_2 + \frac{\omega^2 M_{23}^2}{Z_3}} = \frac{Z_1 Z_2 Z_3 + \omega^2 M_{23}^2 Z_1 + \omega^2 M_{12}^2 Z_3}{Z_2 Z_3 + \omega^2 M_{23}^2}$$

Substituting this variable in the following expression, the S_{11} is proportional to the channel resistance (Z_3) by equation (1),

$$S_{11} = \frac{Z_{in} - R_{source}}{Z_{in} + R_{source}} = \frac{\omega^2 M_{12}^2}{2Z_1 Z_2 + \frac{2\omega^2 M_{23}^2 Z_1}{Z_3} + \omega^2 M_{12}^2}$$

Revised Supplementary Information (page 11):

Supplementary Figure 10 | The simulation result of wireless glucose sensor. The simulated reflection value of glucose concentration from 1 μM to 10 mM.

Comment 6: “Although claimed by the authors, the device may be not suitable for long-term test (up to several hours) of glucose concentration since the enzyme may lose its activity very quickly. Control experiments should be provided to show the long term stability of the glucose sensors.”

Response to Comment 6:

We appreciate for the reviewer’s valuable and insightful comment. As the reviewer suggested, we investigated the long-term stability of our sensors. We stored unused sensors in artificial tear solution for up to 24 hours, and tested their responses to glucose at varied concentrations (Supplementary Fig. 9). We observed no degradation of the sensitivity after 24 hours, which suggests that the enzymes remain active for at least 24 hours. We added this new data to the revised manuscript.

Revised Manuscript (page 7):

We investigated the long-term stability of our sensors with application in a real contact lens in mind. We stored unused sensors in artificial tear solution for up to 24 hours, and tested their responses to glucose at varied concentrations (Supplementary Fig. 9). No degradation of the sensitivity after 24 hours suggests that the enzymes remain active for at least 24 hours. The simple pyrene-chemistry tunes the molecular binding on graphene, and accordingly the multiplexed array of graphene sensors would enable detection of numerous disease-related biomarkers in tear fluid.

Revised Supplementary Information (page 11):

Supplementary Figure 9 | Stability of the glucose sensor. Calibration currents for various glucose concentration with the passage of time.

Comment 7: “How about the influence of salt on the conductivity of graphene in tears? The change of signal before and after being worn on eyes in figure 3e is possibly due to the influence of salt in tears.”

Response to Comment 7:

We would like to thank the reviewer for bringing this point to our attention. As mentioned above (Response to Referee#4, Comment 2), we performed additional experiments to study the effect of the other species in tears. Supplementary Figure 8 below shows the sensor responses to varied concentrations of glucose in buffer (blue) and in artificial tears (red). In both cases, we were able to detect glucose from 1 μM to 10 mM. The slight increase in the baseline current in tears did not degrade the sensitivity. The results confirm that our glucose sensor operates even in the presence of ions and other interfering molecules in tears. We have added the results and discussion to the revised manuscript.

Revised Manuscript (page 7):

The device detected glucose concentration of as low as 1 μM , indicating a 10 times improvement over previously reported contact lens sensors made by evaporated metal electrodes¹⁸. As shown in the calibration curve (Fig. 2d), the sensor was also highly responsive to the typical range of glucose concentrations in tear fluid (0.1 ~ 0.6 mM)¹⁸. Repeating the measurements in artificial tear fluids slightly increased the baseline current but did not degrade the sensitivity (Supplementary Figure 8). The results confirm that our glucose sensor operates even in the presence of ions and other interfering molecules in tears.

Revised Supplementary Information (page 10):

Supplementary Figure 8 | Detection of glucose in different solution environments. The current characteristics of the sensor in response to the glucose dissolved in buffer (blue) and artificial tears (red).

[Additional Modification]

Modification 1:

Franklin Bien has carried out extensive work during the revision process, and we agree to the changed author list.

Revised Manuscript (page 1):

Joohee Kim^{1,†}, Minji Kim^{1,†}, Mi-Sun Lee^{1,†}, Kukjoo Kim¹, Sangyoon Ji¹, Yun-Tae Kim², Jihun Park¹, Kyungmin Na³, Kwi-Hyun Bae⁴, Hong Kyun Kim⁵, Franklin Bien^{3,*}, Chang Young Lee^{2,*}, Jang-Ung Park^{1,*}

Modification 2:

We corrected the Acknowledgements part.

Revised Manuscript (page 1):

This work was supported by the Ministry of Science, ICT & Future Planning and the Ministry of Trade, Industry and Energy (MOTIE) of Korea through the National Research Foundation (2016R1A2B3013592 and 2016R1A5A1009926), the Technology Innovation Program (Grant 10044410), the Nano Material Technology Development Program (2015M3A7B4050308 and 2016M3A7B4910635), the Convergence Technology Development Program for Bionic Arm (NRF-2014M3C1B2048198), the Pioneer Research Center Program (NRF-2014M3C1A3001208), the Human Resource Training Program for Regional Innovation and Creativity (NRF-2014H1C1A1073051). Also, the authors thank CooperVision Awards and financial support by the Development Program of Manufacturing Technology for Flexible Electronics with High Performance (SC0970) funded by the Korea Institute of Machinery and Materials, and by the Development Program of Internet of Nature System (1.150090.01) funded by UNIST.

Reviewer #1 (Remarks to the Author):

The authors do a reasonably good job of responding to reviewer inputs. I should note, however, that the authors' statement that graphene is water impermeable is incorrect. Large area graphene, typical of the type used by the authors, includes grain boundaries and other defects through which water can transport, instantaneously. Numerous studies of this behavior are in the literature. Standard graphene does not work as a hermetic seal. The authors should modify their text accordingly.

Reviewer #2 (Remarks to the Author):

the authors have adequately addressed my requests.

Reviewer #3 (Remarks to the Author):

The authors have reasonably addressed my comments except the 3rd one. Yes, I agree with the authors that it's likely the pressure causes the frequency shift, but my comment was more on the qualitative aspect about the correlation. I suggest the authors measure the capacitance change as a function of applied pressure using a capacitance meter or similar instruments, and then correlate the measured capacitance change with the frequency shift. That would make this work a more convincing. Capacitance-based pressure sensors using silver nanowires have been reported (e.g. *Nanoscale*, 2014, 6, 2345–2352). The authors could perform a similar measurement.

Reviewer #4 (Remarks to the Author):

The revised paper provides additional information for the devices. However, the authors try to avoid some important issues and some questions are answered incorrectly. The major problem of the glucose sensor is the selectivity and sensitivity, which are particularly important for sensing. Since the design of the device is not new as addressed before and device performance is rather poor, I cannot recommend the publication of the paper in *Nature Communications*.

The major problems are as follows:

- (1) The influence of strain on the conductivity of graphene channel is not clarified. It is obvious that CVD graphene will show different conductance under different strain. However, the authors replied that strain will not influence the conductance, which is not true.
- (2) The selectivity of the device should be characterized by adding the major interferences, including ascorbic acid, lactate and uric acid, to see the difference in the signal. On the other hand, the authors should characterize the selectivity of the wireless device shown in Figure 3 instead of graphene FETs.
- (3) We should know the detection limit of the wireless device instead of the graphene FET shown in Figure 2. So the signal to noise ratio in Figure 3c should be provided. I am just wondering why the curves in Figure 3c and 3e show fluctuation and are different from the theoretically simulated curves in Supplementary Figure 10. Is the fluctuation induced by noise? Since the device shows nonlinear response to glucose concentration, the accuracy of the device will be a major problem. It will be very difficult to accurately decide the glucose concentration in tears considering the big fluctuation observed in Figure 3e.
- (4) Since glucose oxidase (GOD) is unstable in solutions, it is necessary to provide explanation why GOD in the devices is stable for 24 hours.
- (5) The authors clearly demonstrated the influence of tears on the conductivity of graphene layer in supplementary Figure 8. So the effect cannot be eliminated in glucose detection. On the other hand, it is necessary to estimate the error induced by other components in tears for glucose detection.

(6) In Figure 3e, why the peak shows no shift before and after wearing it in an eye? Does that mean that the device is not sensitive enough to directly measure the pressure on the eye?

(7) The authors failed to provide reliable results for glucose detection in tears. To prove its correctness, they should compare the data with those obtained by a standard method. In my opinion, the paper just demonstrates the response of the device to glucose and many other factors, but the sensitivity and selectivity of the device are very poor and cannot provide useful information in practical applications.

School of Materials Science and Engineering
Low-Dimensional Carbon Materials Research Center
UNIST-gil 50, Ulsan, Republic of Korea, 689-798

UNIST
ULSAN NATIONAL INSTITUTE OF
SCIENCE AND TECHNOLOGY

Jang-Ung Park

*Associate Professor
School of Materials Science and Engineering,*

+82-52-217-2533
fax: +82-52-217-2509
jangung@unist.ac.kr

November 20, 2016

Dr. Amos Matsiko
Associate Editor
Nature Communications

Re: Decision on manuscript NCOMMS-16-16417

Dear Dr. Amos Matsiko

Thank you for sending us reviewer comments on our manuscript ID NCOMMS-16-16417, “Wearable Smart Sensor Systems Integrated On Soft Contact Lenses For Wireless Ocular Diagnostics” by J. Kim, M. Kim, M.-S. Lee, K. Kim, S. Ji, Y.-T. Kim, J. Park, K. Na, K.-H. Bae, H. K. Kim, F. Bien, C. Y. Lee, and J.-U. Park. Reviewers 1-3 all are satisfied by our responses to their comments, except for a few minor comments that are now addressed, and are strongly supportive of publication of our study in *Nature Communications*. Reviewer 4 had remaining questions and concerns, but we have fully answered them by performing additional experiments and analyses.

We have made careful revisions to our manuscript, *including new data (Supplementary Figure 10, 11 and 13) and revised texts*, to address comments received from all of the reviewers. Point-by-point responses to each of the reviewer reports follow on pages 2-8, and our revised manuscript and supplementary information are also attached. These changes also appear explicitly as red texts in our responses to the comments.

We strongly believe that our response to the reviewers, including revisions to the manuscript / supplementary information, addresses all scientific issues raised in their reports, and that these revisions have provided us with the valuable opportunity to strengthen our work and make the manuscript suitable for publication in *Nature Communications*.

Thank you very much for your time and consideration of our manuscript.

Sincerely,

Jang-Ung Park

[Response to Reviewer #1]

We thank the reviewer for a thoughtful review of our manuscript, and we welcome the opportunity to address and clarify the issues raised in the reviewer report. Our responses are as follows:

Comment 1: “The authors do a reasonably good job of responding to reviewer inputs. I should note, however, that the authors' statement that graphene is water impermeable is incorrect. Large area graphene, typical of the type used by the authors, includes grain boundaries and other defects through which water can transport, instantaneously. Numerous studies of this behavior are in the literature. Standard graphene does not work as a hermetic seal. The authors should modify their text accordingly.”

Response to Comment 1:

We would like to thank the reviewer for bringing this point to our attention. The two-layer passivation using SU8 and graphene effectively reduces permeation of tear fluid for daily wear contacts. However, as the reviewer had suggested, the seal is never perfect, and water can penetrate when the lens is worn for extended periods of time. We therefore modified the manuscript accordingly to address this issue.

Revised Manuscript (page 6):

Here the SU8 as a diffusive barrier lowers molecular concentrations at the already impermeable graphene surface³⁸, ensuring that no damaging molecule from tear fluid reaches the AgNWs. **Grain boundaries in graphene may lower effectiveness of the seal, in particular when the lens is worn for extended periods of time, but the two-layer passivation can provide reasonable protection of the sensor against tear fluids for daily disposable contact lenses.** The formation of AgCl, insoluble salts which could be harmful to the human eye, is also prevented by protecting AgNWs from tear fluid which contains chloride ions.

[Response to Reviewer #2]

“The authors have adequately addressed my requests.”

We thank the reviewer for a positive comment about our manuscript.

[Response to Reviewer #3]

We thank the reviewer for a thoughtful review of and positive comment about our manuscript, and welcome the opportunity to address and clarify the issues raised in the reviewer report. Our responses to the points raised in the report are as follows:

Comment 1: “The authors have reasonably addressed my comments except the 3rd one. Yes, I agree with the authors that it’s likely the pressure causes the frequency shift, but my comment was more on the qualitative aspect about the correlation. I suggest the authors measure the capacitance change as a function of applied pressure using a capacitance meter or similar instruments, and then correlate the measured capacitance change with the frequency shift. That would make this work a more convincing. Capacitance-based pressure sensors using silver nanowires have been reported (e.g. *Nanoscale*, 2014, 6, 2345–2352). The authors could perform a similar measurement.”

Response to Comment 1:

We thank the reviewer for sharing concerns on the intraocular pressure detection. As the reviewer suggested, we performed additional experiments to measure the capacitance of sensor at varying pressures using the LCR meter. As shown in Supplementary Fig. 13 below, we found a high correlation between the pressure and the frequency ($f \sim C^{-0.5}$). We have included these new data in the revised manuscript.

Revised Manuscript (page 10):

In the physiologically relevant range of intraocular pressure, 5 – 50 mmHg¹⁷, the f_{sensor} decreased linearly with pressure by the slope of 2.64 MHz/mmHg (Fig. 4e). **Here the frequency is inversely proportional to the square root of capacitance, $f_{sensor} \sim C^{-0.5}$ (Supplementary Fig. 13).**

Revised Supplementary Information (page 15):

Supplementary Figure 13 | Capacitance response of the intraocular pressure sensor. Normalized capacitance change of the intraocular pressure sensor in the physiological intraocular pressure range (0 - 50 mmHg).

[Response to Reviewer #4]

We thank the reviewer for a thoughtful review of and positive comments about our manuscript, and welcome the opportunity to address and clarify the issues raised in the reviewer report. Our responses to the points raised in the report are as follows:

Comment 1: “The influence of strain on the conductivity of graphene channel is not clarified. It is obvious that CVD graphene will show different conductance under different strain. However, the authors replied that strain will not influence the conductance, which is not true.”

Response to Comment 1:

We thank the reviewer for this comment. The reviewer correctly pointed out that the conductivity of graphene is affected by the strain. CVD-synthesized graphene, however, can be stretched only up to 0.6% in uniaxial tensile strain without damage and cracks in graphene defects or grain boundaries, and the resistance change in this strain range is not significant ($\Delta R < \sim 3\%$)¹, compared to the resistance change in our glucose sensing ($\Delta R > 20\%$ for 0.1mM glucose concentration). Particularly, the resistance change in the glucose concentration range above 0.1 mM is important for diagnosis of diabetes, with consideration of the average glucose level in tear fluid of nondiabetic (0.1-0.6 mM) or diabetic patients (> 0.92 mM)². Furthermore, the graphene-based thin devices exhibit similar electrical properties after transferring them onto non-flat surfaces³ because the thickness is too thin and the bending-induced strain is negligible^{3,4}. Therefore, the influence of strain in the conductance of graphene channel is not dominant for our sensor.

Reference

1. Won, S. *et al.* Double-layer CVD graphene as stretchable transparent electrodes. *Nanoscale* **6**, 6057-6064 (2014).
2. Sen, D. K. & Sarin, G. S. Tear glucose levels in normal people and in diabetic patients. *Br. J. Ophthalmol.* **64**, 693-695 (1980).
3. Park, Y. J., Lee, S.-K., Kim, M.-S., Kim, H. & Ahn, J.-H. Graphene-based conformal devices. *ACS Nano* **8**, 7655-7662 (2014).
4. Kim, D.-H. *et al.* Dissolvable films of silk fibroin for ultrathin conformal bio-integrated electronics. *Nat. Mater.* **9**, 511-517 (2010).

Comment 2: “The selectivity of the device should be characterized by adding the major interferences, including ascorbic acid, lactate and uric acid, to see the difference in the signal. On the other hand, the authors should characterize the selectivity of the wireless device shown in Figure 3 instead of graphene FETs.”

Response to Comment 2:

As the reviewer suggested, we performed additional experiments to show more accurately the selectivity of glucose sensors when interferents (ALU: ascorbic acid, lactate, and uric acid) are present in tears. Here, the concentrations of interfering ALU were chosen to exceed the maximum concentration in tear fluid^{1,2}. Supplementary Figure 11 below shows the sensor responses to the glucose (0.1 mM) dissolved in buffer (black) and ALU solution (A: 50 μ M ascorbic acid, L: 10 mM lactate, U: 10 mM urea) (red). It is clear that the ALU does not affect the signal. The results confirm that our glucose sensor is selective to glucose even in the presence of interfering molecules (ALU) in tears. We have added the results and discussion to the revised manuscript.

Reference

1. Berman, E. R. *Biochemistry of the eye*. New York, pp. 70.
2. Whitehart, D. R. *Biochemistry of the eye*, 2nd ed. Butterworth-Heinemann, Boston, pp.12.

Revised Manuscript (page 8):

The reflection was enlarged at higher glucose concentrations, caused by reduced resistance of the graphene upon glucose binding (Fig. 3c and Supplementary Fig. 10). **The sensor responds specifically to glucose even in the presence of interferents (50 μ M of ascorbic acid, 10 mM of lactate, and 10 mM of uric acid) in the tear (Supplementary Fig. 11).**

Revised Manuscript (page 15):

Selectivity test

For the selectivity test, the sensor is tested by the solution of 0.1 mM glucose, 50 μ M ascorbic acid (Product #PHR1008, Sigma-Aldrich, USA), 10 mM lactate (Product #PHR1113, Sigma-Aldrich, USA), and 10 mM urea (Product #PHR1406, Sigma-Aldrich, USA).

Revised Supplementary Information (page 13):

Supplementary Figure 11 | Selectivity of the glucose sensor. The reflection coefficient of the antenna at the resonance frequency in response to the glucose (0.1 mM) dissolved in buffer (black) and ALU solution (A: 50 μ M ascorbic acid, L: 10 mM lactate, U: 10 mM urea) (red).

Comment 3: “We should know the detection limit of the wireless device instead of the graphene FET shown in Figure 2. So the signal to noise ratio in Figure 3c should be provided. I am just wondering why the curves in Figure 3c and 3e show fluctuation and are different from the theoretically simulated curves in Supplementary Figure 10. Is the fluctuation induced by noise? Since the device shows nonlinear response to glucose concentration, the accuracy of the device will be a major problem. It will be very difficult to accurately decide the glucose concentration in tears considering the big fluctuation observed in Figure 3e.”

Response to Comment 3:

We would like to thank the reviewer for this comment. In our simulation result (Figure 3b), three factors of ideal capacitance, resistance, and inductance are considered. During experimental measurements (Figure 3c and 3e), however, the signal (particularly in GHz ranges) can be affected by environmental factors such as 1) deviation in temperature that related to the thermal efficiency of the antenna¹, 2) fluctuation in atmospheric gases that can cause changes in the moisture content of samples², 3) vibration of the reader coil position in nano-second regime caused by building shaking, elevator, and person’s gait³. These factors can cause fluctuations in our measurements but cannot be simulated with precision. The figure of merit here is the reflection value of the sensor at the resonance frequency. We also calculated the detection limit of the wireless device. The standard deviation of fluctuation in wireless measurements of the antenna-based device is about 0.21, and the limit of detection⁴ (at S/N of 3) is 0.53 μ M. Although some fluctuations in the signal cannot be avoided in experiments, the reflection level is clearly shown at the resonance frequency of 4.1 GHz (Figure 3c) and responded linearly to the glucose concentration (Supplementary Fig. 10).

In Figure 3e, as we already described, the reflection of the wireless glucose sensor increased while the rabbit was wearing the lens, due to the glucose binding in tear fluid of the rabbit. As an example, the glucose concentration in the rabbit’s tear is normally about 0.16mM⁵, which is coincident with the reflection change in Figure 3e.

Reference

1. Proakis, J. G. & Masoud S. Fundamentals of Communication Systems. Upper Saddle River New Jersey: Prentice Hall (2005).
2. Stutzman, W. L. & Thiele, G. A. Antenna theory and design 3rd edition. John Wiley & Sons, Inc. (2013).
3. Skoog, D. A., Holler, F. J. & Crouch, S. R. Principles of instrumental analysis 6th edition. Thomson Brooks/Cole (2007).
4. Shrivastava, A. & Gupta, V. B. Methods for the determination of limit of detection and limit of quantitation of the analytical methods. *Chronicles of Young Scientists* 2, 21-25 (2011).
5. Iguchi, S. *et al.* A flexible and wearable biosensor for tear glucose measurement. *Biomed. Microdevices* 9, 603-609 (2007).

Revised Manuscript (page 8):

The reflection was enlarged at higher glucose concentrations, caused by reduced resistance of the graphene upon glucose binding (Fig. 3c and Supplementary Fig. 10).

Revised Supplementary Information (page 12):
Supplementary Figure 10 | Calibration curve of wireless glucose sensor. The calibration curve generated by reflection coefficient and the glucose concentration from 1 μ M to 10 mM.

Comment 4: “Since glucose oxidase (GOD) is unstable in solutions, it is necessary to provide explanation why GOD in the devices is stable for 24 hours.”

Response to Comment 4:

As the reviewer correctly pointed out, the long-term stability of glucose oxidase (GOD) in solution has been a major issue in developing GOD-based glucose sensors. We have already shown in Supplementary Figure 9 that the GOD in our sensor maintains its activity for 24 hours, and there are several previous studies listed below that support our results. Besteman *et al.* reported that GODs immobilized onto single-walled carbon nanotubes did not show any significant loss of activity for 24 h in water¹. Tsai *et al.* reported the activity of GOD conjugated with single-walled carbon nanotubes maintained for 36 days in solution². Recently Lee *et al.* reported GOD-graphene based glucose sensor in which the GOD was active for 12 days in artificial sweat³. Improving the stability of GODs in some studies required special treatments, but the systems listed above and ours did not require such treatments. The exact mechanism of such long-term stability of GOD on carbon nanotubes and graphene is still unclear, and further studies are necessary to unravel the molecular details, which we believe is beyond the scope of the current work.

Reference

1. Besteman, K., Lee, J. O., Wiertz, F. G., Heering, H. A. & Dekker, C. Enzyme-coated carbon nanotubes as single-molecule biosensors. *Nano Lett.* **3**, 727-730 (2003).
2. Tsai, T. W. *et al.* Adsorption of glucose oxidase onto single-walled carbon nanotubes and its application in layer-by-layer biosensors. *Anal. Chem.* **81**, 7917-7925 (2009).
3. Lee, H. *et al.* A graphene-based electrochemical device with thermoresponsive microneedles for diabetes monitoring and therapy. *Nat. Nanotech.* **11**, 566-572 (2016).

Comment 5: “The authors clearly demonstrated the influence of tears on the conductivity of graphene layer in supplementary Figure 8. So the effect cannot be eliminated in glucose detection. On the other hand, it is necessary to estimate the error induced by other components in tears for glucose detection.”

Response to Comment 5:

As the reviewer pointed out, the ions and other interfering molecular species (ascorbic acid, lactate, and urea) in tear fluid can affect our measurements. However, these interferents does not affect the glucose detection from 1 μ M to 10 mM. Tear fluids change the baseline current slightly but does not degrade the sensitivity to glucose significantly (Supplementary Figure 8). Response to other components such as ascorbic acid, lactate, urea can be ignored in our glucose sensors (Supplementary Figure 10).

Comment 6: “In Figure 3e, why the peak shows no shift before and after wearing it in an eye? Does that mean that the device is not sensitive enough to directly measure the pressure on the eye?”

Response to Comment 6:

As we already described in the manuscript, the contact lens sensor in Figure 3e contains only glucose sensor without the intraocular sensor, so there should be no shift in the frequency.

Comment 7: “The authors failed to provide reliable results for glucose detection in tears. To prove its correctness, they should compare the data with those obtained by a standard method. In my opinion, the paper just demonstrates the response of the device to glucose and many other factors, but the sensitivity and selectivity of the device are very poor and cannot provide useful information in practical applications.”

Response to Comment 7:

Thank you for sharing this concern. The glucose sensor in our study relies on the use of the natural oxygen cosubstrate and generation and detection of hydrogen peroxide, which is a first-generation glucose biosensor. Although the sensors need to be further studied for precise diagnosis, these first-generation sensors can be sufficient for initial diagnosis of diabetes¹. Tear glucose in diabetic patients has been studied for over 80 years since Michail *et al.* first reported². The glucose concentration in tear fluid varies throughout the day and from human eye to eye. Many recent studies, however, have attributed these discrepancies to different methods used for collecting tear fluids such as filter paper or microcapillary. In this regard, the contact lens-based glucose sensor improves reliability of the measurements by precluding the ambiguity during collection of tear fluids. Moreover, several groups have demonstrated validity of the contact lens sensor for monitoring glucose levels, which shows promise of our approach as a home glucose monitor³⁻⁷. We have modified the manuscript to emphasize this point.

Reference

1. Wang, J. Electrochemical glucose biosensors. *Chem. Rev.* **108**, 814-825 (2008)
2. Michail, D., Vancea, P. & Zolog, N. Lacrimal elimination of glucose in diabetic patients. *C. R. Soc. Biol. Paris.* **125**, 194-195 (1937).
3. Alexeev, V. L., Das, S., Finegold, D. N. & Asher, S. A. Photonic crystal glucose-sensing material for noninvasive monitoring of glucose in tear fluid. *Clinical Chemistry* **50**, 2353-2360 (2004).
4. Badugu, R., Lakowicz, J. R. & Geddes, C. D. Ophthalmic glucose monitoring using disposable contact lenses—a review. *Journal of fluorescence* **14**, 617-633 (2004).
5. March, W. F., Mueller, A. & Herbrechtsmeier, P. Clinical trial of a noninvasive contact lens glucose sensor. *Diabetes technology & therapeutics* **6**, 782-789 (2004).
6. Domschke, A., March, W. F., Kabilan, S. & Lowe, C. Initial clinical testing of a holographic non-invasive contact lens glucose sensor. *Diabetes technology & therapeutics* **8**, 89-93 (2006).
7. Ascaso, F. J. & Huerva, V. Noninvasive Continuous Monitoring of Tear Glucose Using Glucose-Sensing Contact Lenses. *Optometry and vision science: official publication of the American Academy of Optometry*.

Revised manuscript (page 7):

The simple pyrene-chemistry tunes the molecular binding on graphene, and accordingly the multiplexed array of graphene sensors would enable detection of numerous disease-related biomarkers in tear fluid. **Although precise diagnosis of glucose may require further development of the sensor, the contact lens sensor can be sufficient for screening prediabetes and daily monitoring of the glucose level.**

Reviewer #1 (Remarks to the Author):

The authors have provided adequate responses to all remaining inputs from reviewers. I feel that the manuscript is suitable for publication in its current form.

Reviewer #3 (Remarks to the Author):

As suggested, the authors have performed additional experiments to measure the capacitance of sensor at varying pressures. The authors added supplementary Figure 13 but did not plot the capacitance change vs. pressure directly; such a relationship is critical. As reported in similar types of pressure sensors including Nature Materials 9, 859–864 (2010), the capacitance should change with the pressure in a linear relationship especially for relatively small pressure. Assuming this is the case in this work (i.e. $C \sim P$) and as the authors correctly pointed out, the resonance frequency shift vs. capacitance change follows $f \sim C^{-0.5}$, the resonance frequency shift vs. the pressure should be $f \sim P^{-0.5}$. However, Figures 4e and f show a linear relationship. The authors should explain.

Reviewer #4 (Remarks to the Author):

I am very pleased to find that the authors have addressed all of my questions correctly and the performance of devices were characterized systematically. The paper can be accepted by Nature Comm.

[Response to Reviewer #1]

“The authors have provided adequate responses to all remaining inputs from reviewers. I feel that the manuscript is suitable for publication in its current form.”

We thank the reviewer for a positive comment about our manuscript.

[Response to Reviewer #3]

We thank the reviewer for a thoughtful review of and positive comment about our manuscript, and welcome the opportunity to address and clarify the issues raised in the reviewer report. Our responses to the points raised in the report are as follows:

Comment 1: “As suggested, the authors have performed additional experiments to measure the capacitance of sensor at varying pressures. The authors added supplementary Figure 13 but did not plot the capacitance change vs. pressure directly; such a relationship is critical. As reported in similar types of pressure sensors including *Nature Materials* 9, 859–864 (2010), the capacitance should change with the pressure in a linear relationship especially for relatively small pressure. Assuming this is the case in this work (i.e. $C \sim P$) and as the authors correctly pointed out, the resonance frequency shift vs. capacitance change follows $f \sim C^{-0.5}$, the resonance frequency shift vs. the pressure should be $f \sim P^{-0.5}$. However, Figures 4e and f show a linear relationship. The authors should explain.”

Response to Comment 1:

We thank the reviewer for this comment. As the reviewer pointed out, we plotted the capacitance change by pressure and included this graph in the revised manuscript (Supplementary Fig. 14). As shown in Supplementary Fig. 14 below, we also found the correlation between the pressure and the capacitance ($C \sim P$).

We performed additional experiments to measure the resonant frequency of our sensor in the wide range of pressure. As shown in Supplementary Fig. 13 below, the graph confirms that a correlation between the pressure and the frequency ($f \sim C^{-0.5}$). The frequency response is, however, almost linear in the low pressure region of 0 - 50 mmHg which corresponds to the intraocular pressure range. Therefore, the measured frequency of Figures 4e and f shifts linearly from the applied pressure. We have added the result and discussion in the revised manuscript.

Revised Manuscript (page 10):

Here the resonance frequency of the sensor is inversely proportional to the square root of pressure, $f_{\text{sensor}} \sim P^{-0.5}$, as shown in Supplementary Fig. 13. In this graph, the frequency response is almost linear for relatively small pressure (below 50 mmHg). In the physiologically relevant range of intraocular pressure, 5 – 50 mmHg¹⁷, the f_{sensor} decreased linearly with pressure by the slope of 2.64 MHz/mmHg (Fig. 4e). Also Supplementary Fig. 14 shows a linear relationship of the relative capacitance change by pressure for this intraocular pressure range ($C \sim P$).

Revised Supplementary Information (page 15):

Supplementary Figure 13 | Frequency response of the pressure sensor using a ball. (a) The resonance frequency change in the wide range of pressure. **(b)** Photograph of the sensor on a ball. Scale bar, 1 cm. In this experiment, a ball was used instead of a bovine eye because this bovine eye is burst in the relatively big pressure regime.

Revised Supplementary Information (page 16):

Supplementary Figure 14 | Capacitance response of the intraocular pressure sensor. (a) Normalized capacitance change of the intraocular pressure sensor in the physiological intraocular pressure range (0 - 50 mmHg). **(b)** Relative capacitance change by the applied intraocular pressure (0 - 50 mmHg).

[Response to Reviewer #4]

“I am very pleased to find that the authors have addressed all of my questions correctly and the performance of devices were characterized systematically. The paper can be accepted by Nature Comm.”

We thank the reviewer for a positive comment about our manuscript.

Reviewer #3 (Remarks to the Author):

The authors have reasonably addressed my comment.